# Iron-deplete diet enhances *Caenorhabditis elegans* lifespan via oxidative stress response pathways

Priyanka Das, Ravi ⓘD & Jogender Singh ⓘD ✉

## Abstract

Gut microbes play a crucial role in modulating host lifespan. However, the microbial factors that influence host longevity and their mechanisms of action remain poorly understood. Using the expression of *Caenorhabditis elegans* FAT-7, a stearoyl-CoA 9-desaturase, as a proxy for lifespan modulation, we conduct a genome-wide bacterial mutant screen and identify 26 *Escherichia coli* mutants that enhance host lifespan. Transcriptomic and biochemical analyses reveal that these mutant diets induce oxidative stress and activate the mitochondrial unfolded protein response (UPRmt). Antioxidant supplementation abolishes lifespan extension, confirming that oxidative stress drives these effects. The extension of lifespan requires the oxidative stress response regulators SKN-1, SEK-1, and HLH-30. Mechanistically, these effects are linked to reduced iron availability, as iron supplementation restores FAT-7 expression, suppresses UPRmt activation, and abolishes lifespan extension. Iron chelation mimics the pro-longevity effects of the mutant diets, highlighting dietary iron as a key modulator of aging. Our findings reveal a bacterial-host metabolic axis that links oxidative stress, iron homeostasis, and longevity in *C. elegans*.

**Keywords** HLH-30; Iron; Lifespan; Oxidative Stress; SKN-1
**Subject Categories** Development; Metabolism; Microbiology, Virology & Host Pathogen Interaction

## Introduction

Aging is a pathophysiological process characterized by the gradual decline of cellular and tissue functions, which significantly increases the risk of age-related disorders, including neurodegenerative diseases, cardiovascular diseases, type 2 diabetes, and cancer (Li et al, 2021). However, like many other biological processes, aging is regulated by canonical signaling pathways and transcription factors, making it amenable to modulation through targeted interventions (Kenyon, 2010). According to the geroscience hypothesis, interventions that extend lifespan may also prevent, delay, and mitigate age-associated disorders (Chmielewski et al, 2024; Kennedy et al, 2014; Li et al, 2021; López-Otín et al, 2013).

Numerous studies have demonstrated that aging can be modulated through genetic, dietary, and pharmacological approaches (Chmielewski et al, 2024; Masoro, 2005; Selman, 2014; Speakman and Mitchell, 2011). For instance, dietary restriction has been shown to extend lifespan and delay the onset of multiple age-related pathologies across various organisms (Chmielewski et al, 2024; Masoro, 2005; Selman, 2014; Speakman and Mitchell, 2011). While ongoing research continues to explore novel therapeutics for aging regulation, there remains a critical need for interventions that are not only effective but also safe, accessible, and practical for everyday implementation.

The gut microbiota, comprising all microorganisms residing in the gastrointestinal tract of an organism, plays a crucial role in maintaining host health and lifespan (Debnath et al, 2021; Rooks and Garrett, 2016; Wang et al, 2024). Its composition changes progressively with age, suggesting that microbiota dysbiosis may represent an additional hallmark of aging (Biagi et al, 2017; Molinero et al, 2023). Indeed, several studies have linked microbial dysbiosis to aging and age-related pathologies (Ragonnaud and Biragyn, 2021). Microbiome-based treatments hold promise due to their potential to modify gut microbe composition through oral interventions (Smith et al, 2017). Moreover, identifying age-modulating metabolites from the microbiome could yield novel strategies for combating aging-related disorders (Gong et al, 2023; Shi et al, 2024).

The nematode *Caenorhabditis elegans* is a widely used model organism in aging research (Kenyon, 2010; Mack et al, 2018). As a bacterivore, *C. elegans* thrives on various bacterial diets, and bacterial metabolites have been shown to influence key life-history traits, including lifespan, making it an excellent system for studying gut microbe-host interactions in aging (M. Feng et al, 2023; Zhang et al, 2017). To date, four distinct genome-wide *Escherichia coli* screens have been conducted to identify bacterial mutants that enhance *C. elegans* lifespan (Han et al, 2017a; Khanna et al, 2016; Shin et al, 2020; Virk et al, 2016). These screens have identified bacterial mutants and metabolites that promote longevity through diverse mechanisms, including dauer formation, activation of the mitochondrial unfolded protein response (UPRmt), and folate limitation (Han et al, 2017a; Khanna et al, 2016; Shin et al, 2020; Virk et al, 2016). Surprisingly, these screens have yielded only minimal overlap in identified mutants (Fig. EV1A), possibly due to poor resolution—often limited to a few time-point measurements—and technical variations, such as differences in liquid versus solid nematode growth media. Nonetheless, these studies suggest that existing screens are far from saturation and that additional *E. coli*

Department of Biological Sciences, Indian Institute of Science Education and Research, Mohali, Punjab 140306, India. ✉E-mail: jogender@iisermohali.ac.in

mutants and mechanisms influencing *C. elegans* lifespan remain to be discovered. They also highlight the need for high-resolution primary screens utilizing phenotypes that serve as proxies for lifespan, followed by secondary screens to validate lifespan changes.

Lipid composition, particularly monounsaturated fatty acid (MUFA) levels, is known to influence lifespan (Schroeder and Brunet, 2015). Long-lived *C. elegans* mutants, including those with reduced insulin-like signaling or dietary restriction mimetics, exhibit elevated MUFA levels (Reis et al, 2011). Δ9 desaturases are key lipogenic enzymes that synthesize MUFAs from saturated fatty acids. *C. elegans* encodes three Δ9 desaturases—FAT-5, FAT-6, and FAT-7 (Brock et al, 2007). Among these, FAT-6 and FAT-7 catalyze the conversion of stearic acid to oleic acid. Studies have shown that dietary MUFA supplementation extends *C. elegans* lifespan, and the expression of Δ9 desaturases is closely linked to aging (Brock et al, 2007; Castillo-Quan et al, 2023; Han et al, 2017b; Reis et al, 2011, 2011; Schroeder and Brunet, 2015). Interestingly, Δ9 desaturase activity is diet-regulated, with diets rich in unsaturated fatty acids repressing its expression (Brock et al, 2007; Choi et al, 1996; Ntambi and Miyazaki, 2003). Given that diet modulates Δ9 desaturase expression and these enzymes are associated with aging, we hypothesized that Δ9 desaturase expression levels could serve as a marker to identify *E. coli* mutants that influence host lifespan.

In this study, we conducted a genome-wide *E. coli* mutant screen to identify microbial factors that modulate *C. elegans* FAT-7 levels. We identified 26 *E. coli* mutants that reduced FAT-7 expression and investigated their effects on host lifespan. Notably, *C. elegans* fed on all 26 *E. coli* mutants exhibited extended lifespan. Transcriptomic profiling indicated that worms experienced oxidative stress on these diets, which was confirmed through biochemical assays. Consistently, we observed activation of the UPRmt in *C. elegans* fed on the mutant *E. coli* strains. Lifespan extension was driven by oxidative stress, as supplementation with the antioxidant N-acetylcysteine (NAC) abolished this effect. Further investigation revealed that iron supplementation reversed all observed phenotypes, including FAT-7 expression, UPRmt activation, and lifespan extension. Conversely, dietary iron limitation recapitulated the effects of the mutant *E. coli* diets, inducing UPRmt activation and lifespan extension. Finally, we demonstrated that the increased lifespan observed under iron-depleted conditions was mediated by genetic pathways associated with oxidative stress responses, including the nuclear factor erythroid 2-related factor SKN-1, the MAP kinase kinase SEK-1, and the TFEB ortholog HLH-30. Our findings uncovered a metabolic interaction between bacteria and the host that connects oxidative stress, iron homeostasis, and longevity in *C. elegans*.

# Results

## Genome-wide bacterial screen identifies *E. coli* mutants that modulate *C. elegans* FAT-7 levels

To identify *E. coli* mutants that enhance the lifespan of *C. elegans*, we used changes in *C. elegans* FAT-7 expression as a proxy for lifespan alterations. We designed a genome-wide screen using the Keio collection to identify *E. coli* mutants that modify *C. elegans* FAT-7 expression, utilizing the *fat-7p::fat-7::GFP* reporter strain.

Mutants that alter FAT-7 expression would subsequently be tested for their effects on *C. elegans* lifespan in a secondary screen (Fig. 1A). The Keio collection consists of single-gene deletion mutants in 3985 genes of the *E. coli* BW25113 strain (Baba et al, 2006). Synchronized L1 larvae of the FAT-7 reporter strain were fed individual *E. coli* mutants seeded on nematode growth medium (NGM) plates and allowed to develop into gravid adults. Plates were then screened under a fluorescence stereomicroscope to identify mutants that caused either an increase or a decrease in green fluorescence protein (GFP) expression compared to the wild-type *E. coli* BW25113 strain (Fig. 1A).

From this primary screen, we identified 26 *E. coli* mutants that significantly reduced FAT-7::GFP levels (Figs. 1B and EV1B; Table EV1). However, no mutants were found that significantly increased FAT-7::GFP expression. For clarity, we will refer to the *E. coli* mutants that suppressed FAT-7 expression in *C. elegans* as FAT-7-suppressing diets, while the BW25113 strain will be referred to as the control diet. Notably, worms fed on all the FAT-7-suppressing diets exhibited delayed development compared to those grown on the control diet (Fig. 1C).

Gene ontology (GO) analysis for the molecular function of the identified bacterial mutants revealed enrichment of categories related to electron transport and oxidoreductase activity (Fig. EV1C). This suggested that the *E. coli* mutants that suppress FAT-7::GFP might have a disrupted redox balance.

## *E. coli* mutants that decrease FAT-7 levels extend *C. elegans* lifespan

Next, we tested whether the *E. coli* mutants that reduce FAT-7 levels also impact the lifespan of *C. elegans*. Notably, all FAT-7-suppressing diets increased the mean survival of worms compared to those grown on the control diet (Figs. 2A–D and EV2; Dataset EV1). Comparing these hits with *E. coli* mutants identified in previous lifespan-enhancing screens revealed no overlap with earlier findings (Appendix Fig. S1). These results suggest that FAT-7 expression serves as a reliable proxy for identifying diets that promote longevity.

The identified hits were associated with diverse pathways. To investigate the mechanisms underlying their effects, we selected four mutant diets linked to distinct metabolic processes: Δ*tktA*, Δ*yciA*, Δ*pdeI*, and Δ*allD*. *tktA* encodes the primary transketolase enzyme in *E. coli* that serves as a reversible link between glycolysis and the pentose phosphate pathway. *yciA* encodes an acyl-CoA thioesterase, *pdeI* encodes a predicted c-di-GMP-specific phosphodiesterase, and *allD* encodes ureidoglycolate dehydrogenase, involved in allantoin assimilation. Worms fed these four mutant diets displayed enhanced lifespans compared to those on the control diet (Fig. 3A). These bacterial mutants did not exhibit any growth defects compared to the wild-type control (Appendix Fig. S2).

FAT-7 converts stearic acid to oleic acid, and elevated oleic acid levels are known to suppress *fat-7* expression (Venkatesh et al, 2023). We hypothesized that bacterial diets might reduce FAT-7 expression because they have elevated levels of oleic acid in *C. elegans*. Previous studies have shown that oleic acid supplementation extends *C. elegans* lifespan (Han et al, 2017b), suggesting that lifespan extension on FAT-7-suppressing diets might result from increased oleic acid levels. FAT-2 encodes a Δ12 desaturase, and

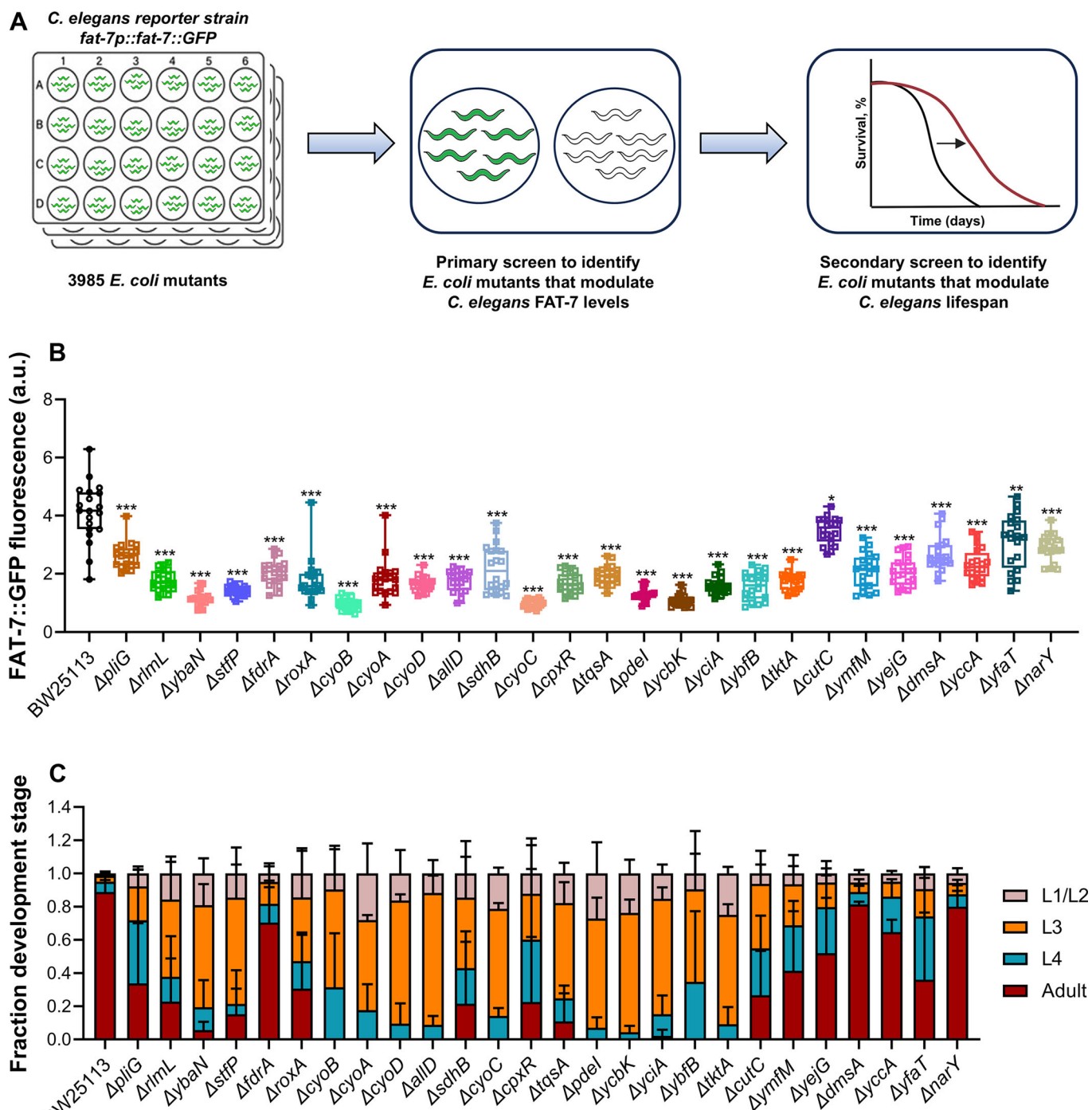

**Figure 1. Genome-wide bacterial screen identifies *E. coli* mutants that modulate *C. elegans* FAT-7 levels.**

(A) Schematic representation of the genome-wide primary bacterial screen used to identify *E. coli* mutants that modulate *C. elegans* FAT-7 levels, followed by a secondary screen to assess their impact on lifespan. (B) Quantification of GFP levels of *fat-7p::fat-7::GFP* worms grown on *E. coli* BW25113 and mutant diets. ***$P < 0.0001$ compared to the control on all the mutant diets except for *yfaT* ($P = 0.0025$), and *cutC* ($P = 0.0189$) via the *t* test ($n = 20$ worms each from two independent experiments). In the boxplots, the central bands represent the median value, the boxes represent the upper and lower quartiles, and the whiskers represent the minimum and maximum values. (C) Quantification of different developmental stages of N2 worms grown on *E. coli* BW25113 and mutant diets at 20 °C, 60 h after transferring synchronized L1 larvae ($n = 3$ biological replicates; animals per condition per replicate >45). Data represent the mean and standard deviation from three independent experiments. Source data are available online for this figure.

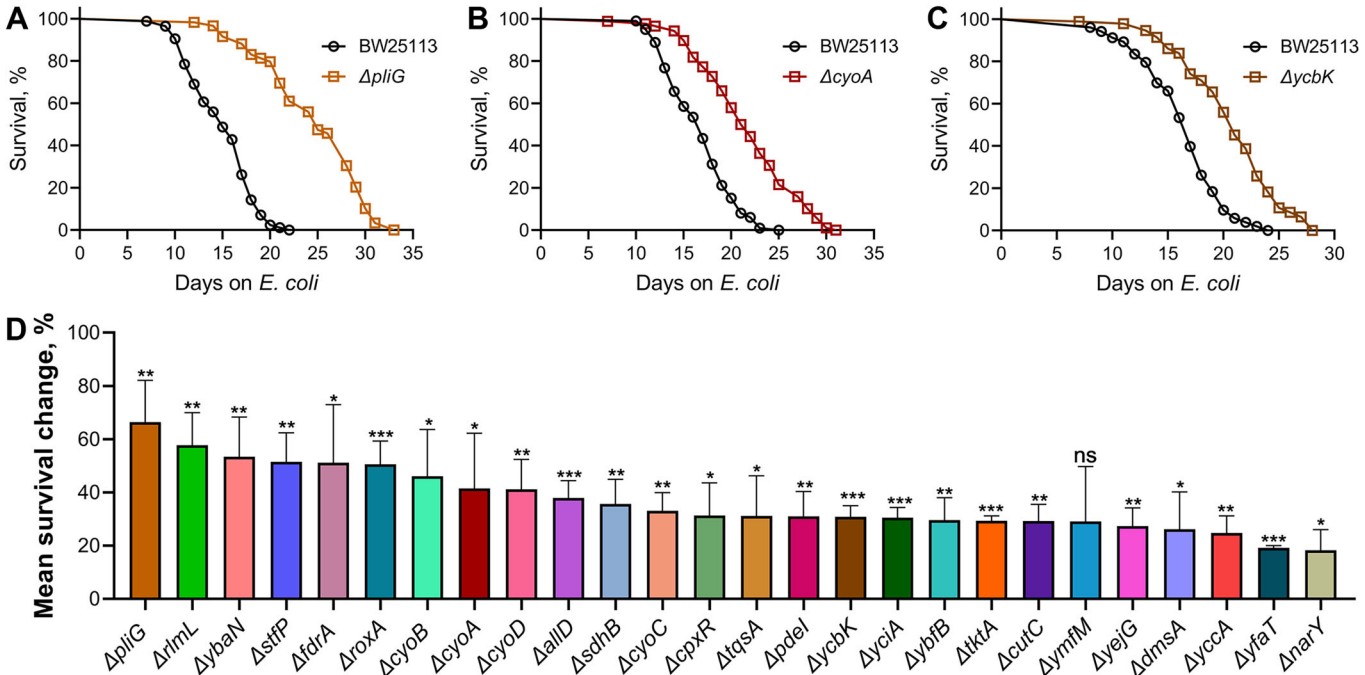

**Figure 2. *E. coli* mutants that decrease FAT-7 levels extend *C. elegans* lifespan.**

(A–C) Representative survival curves of N2 worms fed on *E. coli* mutants *ΔpliG* (A), *ΔcyoA* (B), and *ΔycbK* (C), along with the BW25113 controls. *P* < 0.001 for all the plots (*n* = 3 biological replicates; animals per condition per replicate >58). (D) The percent change in mean survival of N2 worms fed on *E. coli* mutant diets relative to the BW25113 control. *P* values compared to the control calculated via the *t* test are the following: *pliG* = 0.0018, *rlmL* = 0.0012, *ybaN* = 0.0034, *stfP* = 0.0012, *fdrA* = 0.0155, *roxA* = 0.0006, *cyoB* = 0.0105, *cyoA* = 0.0256, *cyoD* = 0.0032, *allD* = 0.0005, *sdhB* = 0.0026, *cyoC* = 0.0011, *cpxR* = 0.0114, *tqsA* = 0.0234, *pdeI* = 0.0045, *ycbA* = 0.0002, *yciA* = 0.0002, *ybfB* = 0.0036, *tktA* < 0.0001, *cutC* = 0.0012, *ymfm* = 0.0708, *yejG* = 0.0023, *dmsA* = 0.0310, *yccA* = 0.0026, *yfaT* < 0.0001, and *narY* = 0.0151. Data represent the mean and standard deviation from three independent experiments. Source data are available online for this figure.

mutants lacking *fat-2* cannot convert oleic acid into linoleic acid, leading to elevated oleic acid levels (Watts and Browse, 2002). To examine whether oleic acid accumulation accounts for the extended lifespan observed with FAT-7-suppressing diets, we studied the survival of *fat-2(wa17)* hypomorphic mutants on the four selected diets. Surprisingly, *fat-2(wa17)* animals exhibited an increased lifespan on these diets (Fig. 3B), suggesting that the lifespan extension is unlikely to be due to oleic acid accumulation in *C. elegans* fed the FAT-7-suppressing diets.

To further investigate the role of oleic acid in lifespan extension on FAT-7-suppressing diets, we examined the effects of oleic acid supplementation. As expected, oleic acid supplementation increased the lifespan of worms on the control diet (Fig. EV3A–D). Oleic acid also extended lifespan on the FAT-7-suppressing diets, with variable effects on each of the mutant diets. For example, oleic acid supplementation had a neutral effect on lifespan on *tktA* and *allD* mutant diets compared to its effect on the control diet. On the other hand, while oleic acid supplementation had a negative effect on lifespan on the *yciA* diet compared to its effect on the control diet, it had a positive effect on the *pdeI* mutant diet (Fig. EV3A–D). Overall, these results suggested that the observed lifespan extension under these conditions is unlikely to be driven by oleic acid.

We next asked whether downregulation of FAT-7 itself was responsible for the extended lifespan on these diets. To test this, we overexpressed FAT-7 to determine whether this manipulation could reverse the lifespan extension seen on the mutant diets. A

previous study reported that intestinal overexpression of FAT-7 extends *C. elegans* lifespan (Han et al, 2017b). Moreover, FAT-7 expression is primarily observed in the intestine in the FAT-7::GFP reporter strain. Therefore, we overexpressed FAT-7 under an intestine-specific promoter. Unexpectedly, we did not observe increased lifespan upon FAT-7 overexpression in worms fed the control diet (Fig. EV3E–H). This discrepancy from Han et al, 2017b may reflect differences in experimental conditions, such as bacterial diets or transgene expression levels. Nonetheless, intestinal FAT-7 overexpression only partially reduced lifespan extension on the FAT-7-suppressing diets (Fig. EV3E–H), indicating that suppression of FAT-7 expression contributes only modestly to the observed phenotype. Collectively, these results suggested that, within our experimental framework, FAT-7 expression likely functions as an indirect proxy for lifespan regulation rather than a direct determinant.

## Mutant *E. coli* diets induce oxidative stress in *C. elegans*

To investigate the mechanisms underlying the increased lifespan of *C. elegans* on mutant *E. coli* diets, we examined transcriptomic changes in worms fed these diets. Wild-type worms were grown on the control diet and four mutant diets (*ΔtktA*, *ΔpdeI*, *ΔyciA*, and *ΔallD*) until day 1 of adulthood, followed by RNA sequencing. Comparative analysis revealed that 1281 upregulated genes were shared across worms fed all four mutant diets (Fig. 3C;

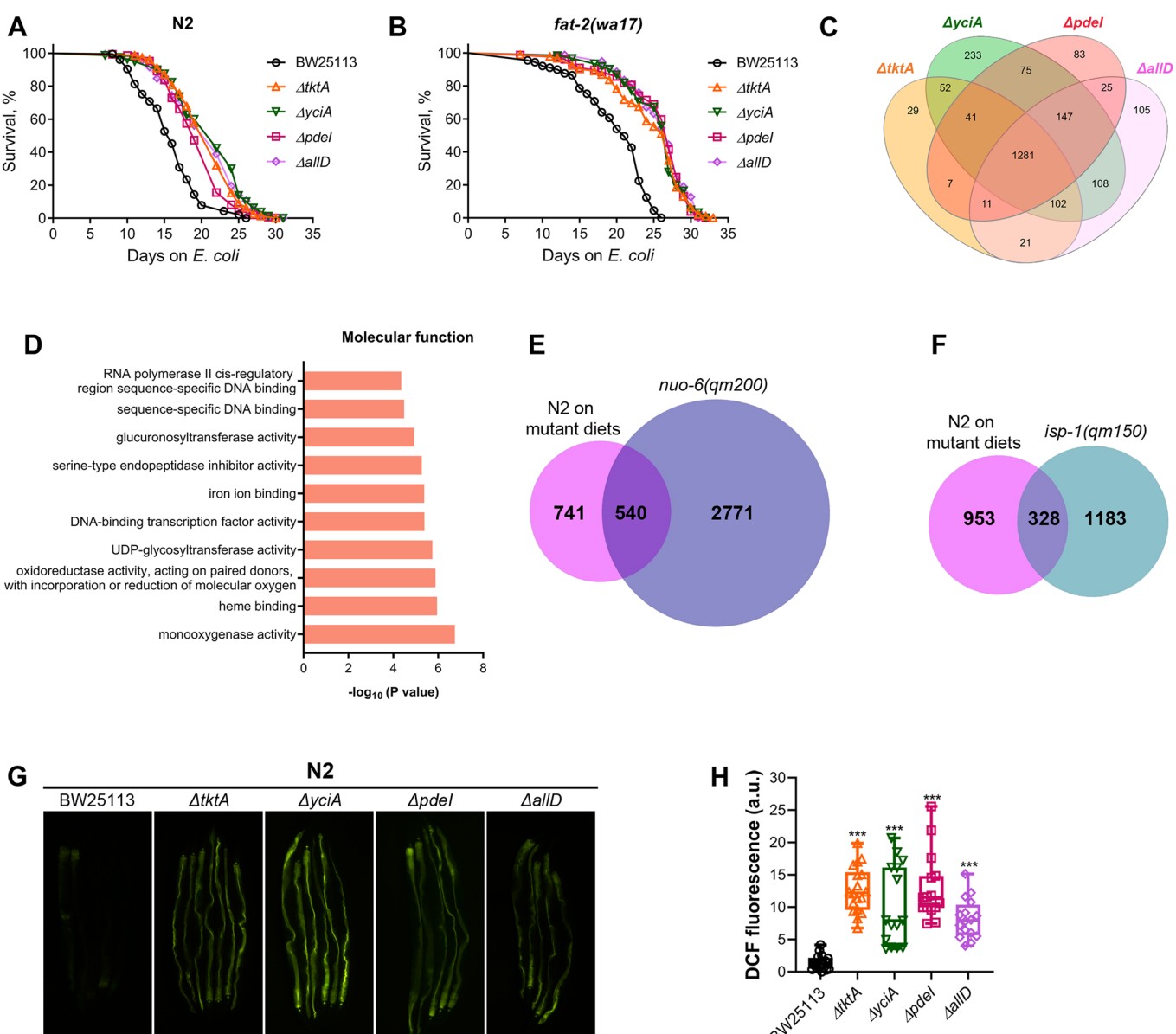

**Figure 3. Mutant *E. coli* diets induce oxidative stress in *C. elegans*.**

(A) Representative survival curves of N2 worms fed on *ΔtktA*, *ΔyciA*, *Δpdel*, and *ΔallD E. coli* mutants along with the BW25113 control. *P* < 0.001 for all the mutant diets compared to the BW25113 control (*n* = 3 biological replicates; animals per condition per replicate >75). (B) Representative survival curves of *fat-2(wa17)* worms fed on *ΔtktA*, *ΔyciA*, *Δpdel*, and *ΔallD E. coli* mutants along with the BW25113 control. *P* < 0.001 for all the mutant diets compared to the BW25113 control (*n* = 3 biological replicates; animals per condition per replicate >60). (C) Venn diagram showing the overlap of upregulated genes in N2 worms fed on *ΔtktA*, *ΔyciA*, *Δpdel*, and *ΔallD E. coli* mutants compared to the BW25113 control. (D) Gene ontology enrichment analysis of molecular function for the common 1281 genes upregulated in N2 worms grown on *E. coli* mutants *ΔtktA*, *ΔyciA*, *Δpdel*, and *ΔallD*. The statistical analysis was performed using Fisher's exact test. (E) Venn diagram showing the overlap between genes upregulated in N2 worms fed the FAT-7-suppressing diets and upregulated in *nuo-6(qm200)* worms (Senchuk et al, 2018). The overlap exhibits an enrichment factor of 2.6. The *P* value for the overlap is 5.83 × 10$^{-117}$ (hypergeometric test). (F) Venn diagram showing the overlap between genes upregulated in N2 worms fed the FAT-7-suppressing diets and upregulated in *isp-1(qm150)* worms (Senchuk et al, 2018). The overlap exhibits an enrichment factor of 3.39. The *P* value for the overlap is 1.67 × 10$^{-95}$ (hypergeometric test). (G) Representative fluorescence images of N2 worms grown on *ΔtktA*, *ΔyciA*, *Δpdel*, and *ΔallD E. coli* mutants, along with the BW25113 control, and exposed to 2',7'-dichlorofluorescein diacetate for 5 h before imaging. Scale bar = 200 μm. (H) Quantification of fluorescence levels of 2',7'-dichlorofluorescein (DCF) in N2 worms grown on *ΔtktA*, *ΔyciA*, *Δpdel*, and *ΔallD E. coli* mutants, along with the BW25113 control, and exposed to 2', 7'-dichlorodihydrofluoroscein diacetate for 5 h before imaging. ***P* < 0.0001 via the *t* test (*n* = 15–18 worms each). In the boxplots, the central bands represent the median value, the boxes represent the upper and lower quartiles, and the whiskers represent the minimum and maximum values. Source data are available online for this figure.

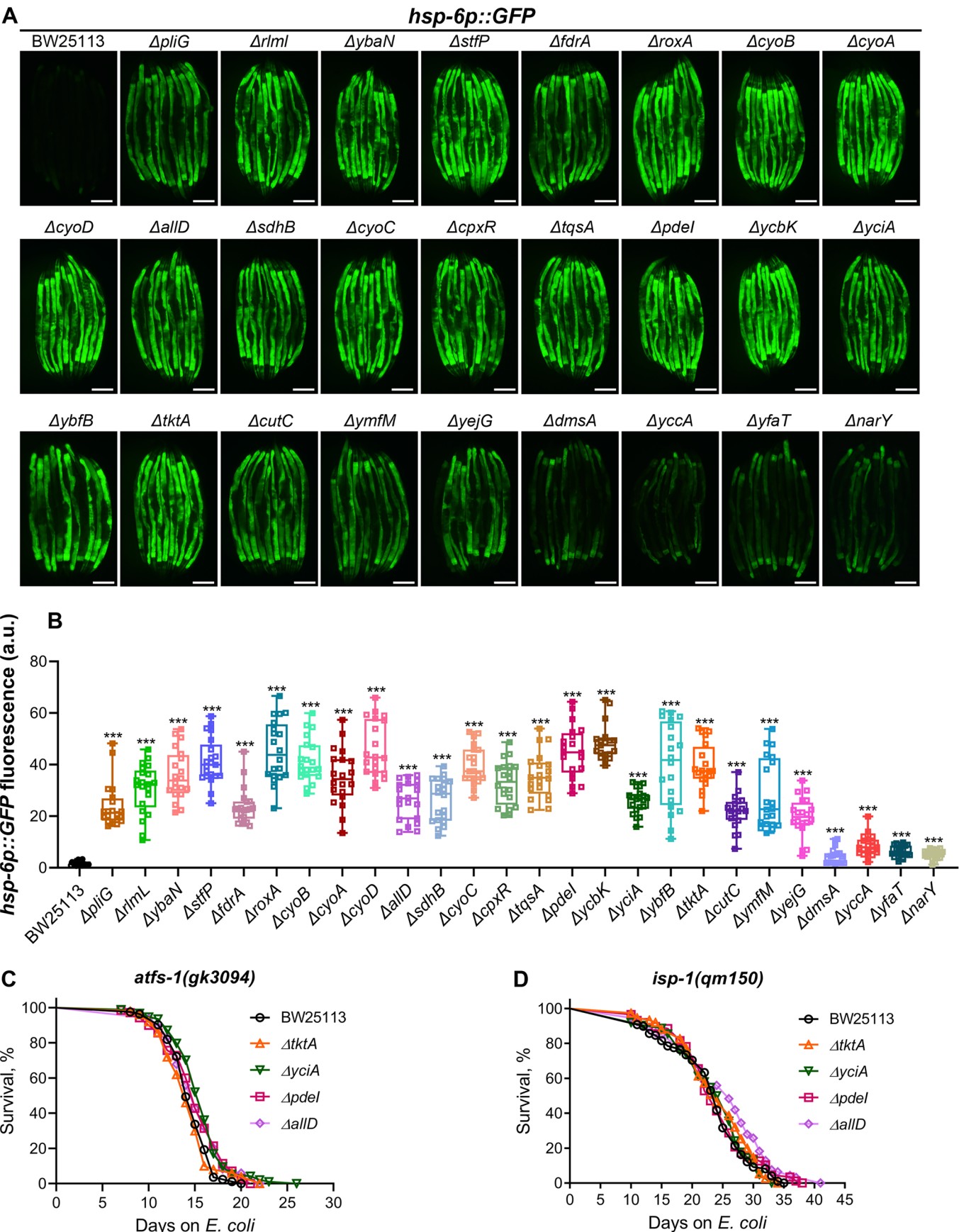

**A** *hsp-6p::GFP*

**C** *atfs-1(gk3094)*

**D** *isp-1(qm150)*

◄ **Figure 4.   Mutant *E. coli* diets that suppress *C. elegans* FAT-7 activate host mitochondrial UPR.**

(A) Representative fluorescence images of *hsp-6p::GFP* worms grown on BW25113 and mutant *E. coli* diets. Scale bar = 200 µm. (B) Quantification of GFP levels of *hsp-6p::GFP* worms grown on BW25113 and mutant *E. coli* diets. $P < 0.0001$ compared to the control on all the mutant diets except *dmsA* ($P = 0.0006$) via the *t* test ($n = 19$–21 worms each). In the boxplots, the central bands represent the median value, the boxes represent the upper and lower quartiles, and the whiskers represent the minimum and maximum values. (C) Representative survival curves of *atfs-1(gk3094)* worms fed on Δ*tktA*, Δ*yciA*, Δ*pdeI*, and Δ*allD E. coli* mutants along with the BW25113 control. $P < 0.001$ for Δ*yciA*, $P < 0.05$ for Δ*pdeI* and Δ*allD*, and nonsignificant for Δ*tktA* compared to the BW25113 control ($n = 3$ biological replicates; animals per condition per replicate >49). (D) Representative survival curves of *isp-1(qm150)* worms fed on Δ*tktA*, Δ*yciA*, Δ*pdeI*, and Δ*allD E. coli* mutants along with the BW25113 control. $P < 0.01$ for Δ*allD* and nonsignificant for Δ*tktA*, Δ*yciA*, and Δ*pdeI* compared to the BW25113 control ($n = 3$ biological replicates; animals per condition per replicate >51). Source data are available online for this figure.

Dataset EV2). Similarly, a significant overlap was observed among downregulated genes (Appendix Fig. S3A; Dataset EV3). These findings suggested a shared molecular mechanism underlying the lifespan-enhancing effects of these diets.

GO analysis of the molecular functions associated with the downregulated genes on all mutant diets revealed enrichment for nucleic acid binding and protein heterodimerization activities (Appendix Fig. S3B). On the other hand, GO analysis of the 1,281 genes upregulated on all mutant diets showed enrichment for molecular functions related to monooxygenase, oxidoreductase, UDP-glucosyltransferase, and iron ion binding activities (Fig. 3D). Notably, these genes are linked to detoxification pathways and are typically upregulated in response to oxidative stress. This suggested that worms feeding on mutant diets may experience elevated reactive oxygen species (ROS) levels compared to those on the control diet. Mutations in the mitochondrial genes *nuo-6* and *isp-1*, which encode subunits of complex I and III of the mitochondrial respiratory chain, respectively, are known to increase superoxide levels (Yang and Hekimi, 2010). A comparison of the genes upregulated on FAT-7-suppressing diets with those induced in *nuo-6* and *isp-1* partial loss-of-function mutants revealed significant overlap (Fig. 3E,F), supporting the notion that worms fed on FAT-7-suppressing diets experience elevated ROS. Direct measurements confirmed this prediction, showing significantly higher ROS levels in worms fed FAT-7-suppressing diets relative to the control diet (Fig. 3G,H).

## Mutant *E. coli* diets that suppress *C. elegans* FAT-7 activate host mitochondrial UPR

Mitochondria are highly sensitive to elevated ROS, which can create a proteotoxic environment and disrupt protein trafficking across the inner mitochondrial membrane (Melber and Haynes, 2018). Such disruptions can impair mitochondrial protein import and activate the UPRmt, a conserved pathway that restores mitochondrial homeostasis (Melber and Haynes, 2018; Shpilka and Haynes, 2018). Activation of the UPRmt has been associated with lifespan extension in *C. elegans* (Bennett et al, 2014; Shpilka and Haynes, 2018; Xin et al, 2022). Consistently, the mitochondrial mutants *nuo-6* and *isp-1*, which exhibit significant transcriptomic overlap with worms fed FAT-7-suppressing diets, also activate the UPRmt, and their lifespan extension depends on this pathway (Wu et al, 2018).

Our ROS measurement analysis suggested elevated ROS levels in worms fed FAT-7-suppressing diets. To determine whether UPRmt was activated in worms fed these diets, we examined the expression of *hsp-6*, a mitochondrial chaperone and reporter for UPRmt activation (Yoneda et al, 2004). We grew *hsp-6p::GFP* worms on the

FAT-7-suppressing diets until the day-1-adult stage. GFP fluorescence levels were significantly increased in worms fed mutant diets compared to the control diet, indicating UPRmt activation (Fig. 4A,B). While the UPRmt activation and lifespan extension were observed on all 26 mutant diets, there was no strong correlation between *hsp-6p::GFP* expression levels and lifespan extension (Appendix Fig. S4A).

Next, we investigated whether UPRmt activation is required for the lifespan extension observed on the mutant diets. ATFS-1, a key transcription factor, contains both a mitochondrial localization signal and a weak nuclear localization signal. Under normal conditions, ATFS-1 is imported into mitochondria and degraded. However, during mitochondrial dysfunction, its import is blocked, and ATFS-1 translocates to the nucleus to activate the UPRmt (Nargund et al, 2012). We analyzed the survival of the ATFS-1 loss-of-function mutant *atfs-1(gk3094)* on the FAT-7-suppressing diets. The lifespan extension observed on these diets was abolished in *atfs-1(gk3094)* animals (Fig. 4C). Because *isp-1(qm150)* mutants also activate the UPRmt and display extended lifespan (Wu et al, 2018), we asked whether their longevity pathway overlapped with that induced by FAT-7-suppressing diets. Indeed, lifespan extension was abolished in *isp-1(qm150)* mutants fed these diets (Fig. 4D). These findings demonstrated that UPRmt activation is essential for the lifespan-enhancing effects of mutant diets in *C. elegans*.

Given that the transcriptional profiles of worms fed FAT-7-suppressing diets significantly overlapped with *nuo-6* and *isp-1* loss-of-function mutants, we next asked whether these mitochondrial mutants also showed reduced *fat-7* expression. Indeed, transcriptomic data from multiple studies showed that *fat-7* is consistently downregulated in *nuo-6* and *isp-1* mutants (Park et al, 2020; Senchuk et al, 2018; Wu et al, 2018; Yee et al, 2014). This led us to hypothesize that mitochondrial stress more broadly downregulates *fat-7*. Supporting this, reanalysis of published datasets revealed reduced *fat-7* expression in several mitochondrial mutants with activated UPRmt, including *clk-1, cco-1*, and *hsp-6* (Fischer et al, 2014; Mao et al, 2019; Matilainen et al, 2017; Tian et al, 2016; Zhu et al, 2020). To confirm whether mitochondrial stress results in the downregulation of *fat-7*, we exposed the *fat-7p::fat-7::GFP* reporter strain to paraquat (PQ). While PQ treatment resulted in the upregulation of *hsp-6p::GFP*, it led to the downregulation of FAT-7::GFP levels (Fig. EV4A–D). Similarly, knockdown of *tomm-22*, which elicits UPRmt, also led to downregulation of FAT-7::GFP (Fig. EV4E–H). Together, these findings suggested that mitochondrial stress suppresses *fat-7* expression and that the FAT-7 reporter may have functioned as an indirect indicator of mitochondrial stress in our Keio library screen.

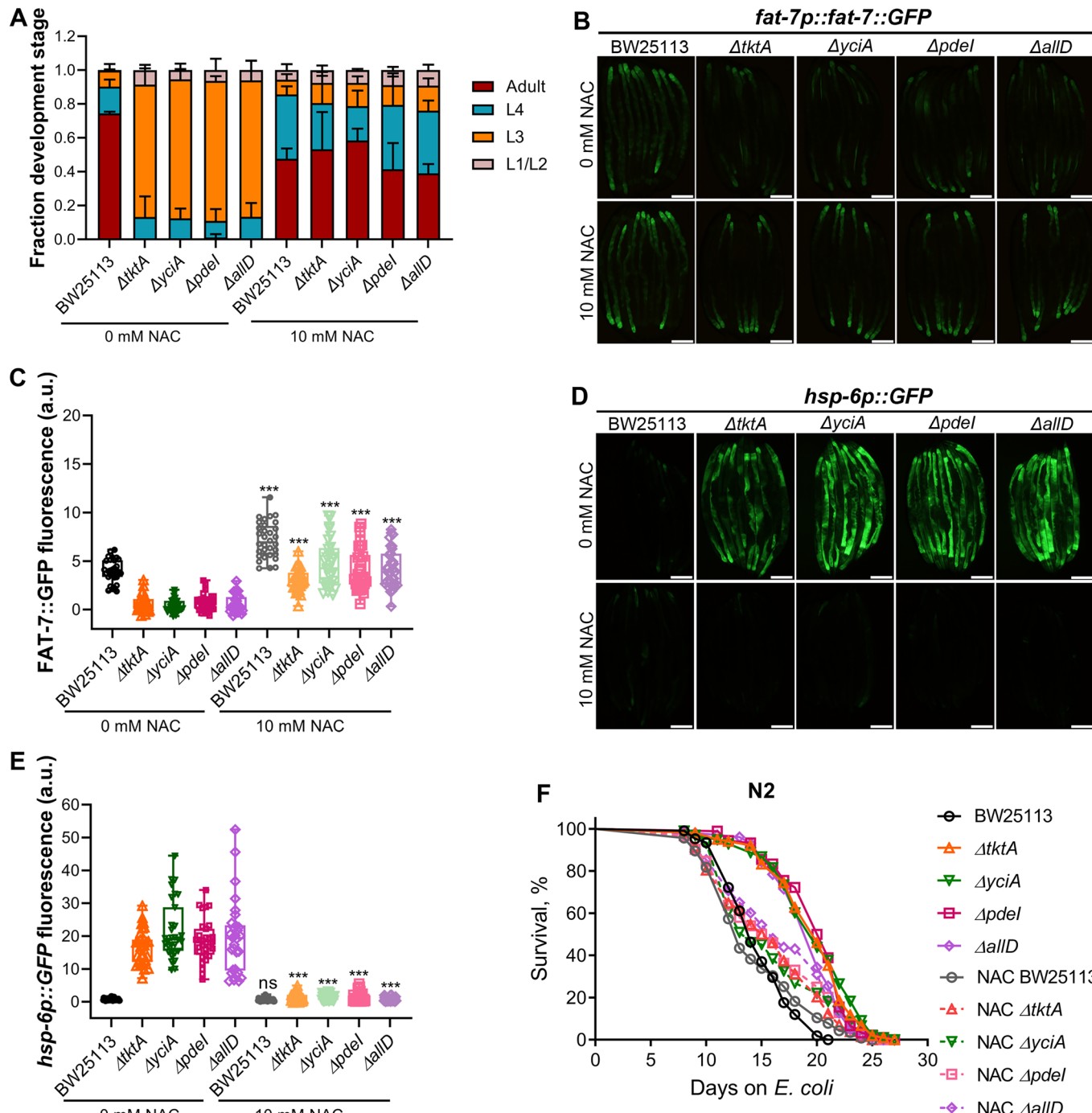

## Antioxidant supplementation rescues mutant diet-induced phenotypes

Because worms exhibited elevated ROS on the FAT-7-suppressing diets, we asked whether the associated phenotypes were driven by increased ROS levels. To test this, we supplemented the diets with the antioxidant NAC and examined the resulting phenotypes. NAC supplementation restored normal development in worms grown on FAT-7-suppressing diets (Fig. 5A). It also nearly completely rescued FAT-7::GFP expression (Fig. 5B,C) and significantly reduced *hsp-6* expression (Fig. 5D,E).

Importantly, NAC supplementation abolished the lifespan extension normally observed on FAT-7-suppressing diets (Fig. 5F). Together, these findings indicated that the phenotypes induced by FAT-7-suppressing diets are primarily mediated by elevated ROS.

## Iron supplementation rescues mutant diet-induced phenotypes

A previous study by Zhang et al identified *E. coli* Keio mutants that delayed *C. elegans* development (Zhang et al, 2019). Interestingly, most

◀  **Figure 5.   N-acetylcysteine (NAC) supplementation rescues mutant diet-induced phenotypes.**

(A) Quantification of different developmental stages of N2 worms grown at 20 °C for 60 h after transferring synchronized L1 larvae onto *E. coli* BW25113 and FAT-7-suppressing diets supplemented with 0 or 10 mM NAC (*n* = 3 biological replicates; animals per condition per replicate >53). Data represent the mean and standard deviation from three independent experiments. (B) Representative fluorescence images of *fat-7p::fat-7::GFP* worms grown on BW25113 and FAT-7-suppressing diets supplemented with 0 or 10 mM NAC. Scale bar = 200 μm. (C) Quantification of GFP levels of *fat-7p::fat-7::GFP* worms grown on BW25113 and FAT-7-suppressing diets supplemented with 0 or 10 mM NAC. *P* values were calculated by comparing NAC-supplemented diets (10 mM) to their respective unsupplemented controls (0 mM NAC). For all the comparisons $P < 0.0001$ via the *t* test. (*n* = 29–32 worms each). (D) Representative fluorescence images of *hsp-6p::GFP* worms grown on BW25113 and FAT-7-suppressing diets supplemented with 0 or 10 mM NAC. Scale bar = 200 μm. (E) Quantification of GFP levels of *hsp-6p::GFP* worms grown on BW25113 and FAT-7-suppressing diets supplemented with 0 or 10 mM NAC. *P* values were calculated by comparing NAC-supplemented diets (10 mM) to their respective unsupplemented controls (0 mM NAC). For BW25113, $P = 0.5008$, and for all other diets, $P < 0.0001$ via the *t* test. ns, nonsignificant (*n* = 30–32 worms each). (F) Representative survival curves of N2 worms fed on Δ*tktA*, Δ*yciA*, Δ*pdel*, and Δ*allD E. coli* mutants, along with the BW25113 control, supplemented with 0 or 10 mM NAC. For 0 mM NAC, $P < 0.001$ for Δ*tktA*, Δ*yciA*, Δ*pdel*, and Δ*allD* compared to their BW25113 control. For 10 mM NAC, $P < 0.05$ for Δ*tktA* and Δ*yciA*, $P < 0.01$ for Δ*pdel*, and $P < 0.001$ for Δ*allD* compared to their BW25113 control (*n* = 3 biological replicates; animals per condition per replicate >66). Data information: In the boxplots in (C, E), the central bands represent the median value, the boxes represent the upper and lower quartiles, and the whiskers represent the minimum and maximum values. Source data are available online for this figure.

of these mutants also upregulated *hsp-6p::GFP* expression in *C. elegans* that were rescued by NAC supplementation. Similarly, our study observed delayed development and increased *hsp-6* expression in worms fed on mutant diets. A comparison between the two studies revealed a nearly complete overlap, with 23 out of 26 mutants from our screen matching those identified by Zhang et al (Appendix Fig. S4B). Zhang et al attributed the observed phenotypes to elevated ROS levels and reduced bioavailable iron in *E. coli* mutants.

Based on these findings, we hypothesized that the phenotypes observed on FAT-7-suppressing diets could be due to low bioavailable iron. Indeed, supplementing these diets with ferric chloride restored worm development and *hsp-6p::GFP* expression to levels observed on the control diet (Fig. 6A–C; Appendix Fig. S4C). Ferric chloride supplementation also rescued the reduced FAT-7::GFP expression levels to those seen on the control diet (Fig. 6D,E). Moreover, the pro-longevity effects of mutant diets were abolished with ferric chloride supplementation (Fig. 6F). Taken together, these data demonstrated that iron supplementation rescues all phenotypes associated with the mutant diets.

### Low dietary iron mimics mutant diet-induced phenotypes

To determine whether low-iron levels directly cause the observed phenotypes, we tested the effects of the iron chelator 2,2'-bipyridyl. Worms fed on *E. coli* grown with bipyridyl exhibited delayed development (Fig. 7A), increased *hsp-6p::GFP* expression (Fig. 7B,C), and reduced FAT-7::GFP expression (Fig. 7D,E). Importantly, bipyridyl supplementation extended *C. elegans* lifespan (Fig. 7F). These findings showed that iron chelation mimics all the effects of the mutant diets.

To explore whether the lifespan extension observed under iron chelation and on mutant diets involved overlapping mechanisms, we studied the survival of worms fed on mutant diets upon supplementation with bipyridyl. While supplementation of bipyridyl enhanced *C. elegans* lifespan on the control diet (Fig. 7F), it did not further extend the lifespan on mutant diets (Fig. 7G–K). Taken together, these results suggested that the lifespan extension induced by mutant diets arises from a low-iron environment, similar to that created by bipyridyl supplementation.

### Lifespan extension under low dietary iron depends on oxidative stress response pathways

We next investigated the mechanisms underlying lifespan extension in worms fed FAT-7-suppressing or iron-depleted diets. Because

changes in food intake can influence lifespan, we first tested whether the mutant diets affected feeding behavior. Worms fed FAT-7-suppressing diets showed a significant reduction in pharyngeal pumping (Fig. EV5A). To determine whether reduced food intake accounted for the observed phenotypes, we examined *eat-2(ad465)* mutants, which display markedly reduced pharyngeal pumping (Avery, 1993). If decreased pumping were causal, *eat-2* mutants should exhibit reduced *fat-7* expression and elevated *hsp-6* expression. However, *eat-2* mutants showed neither phenotype (Fig. EV5B–E). Thus, reduced pharyngeal pumping was not the cause of the observed phenotypes but was more likely a consequence of elevated oxidative stress. Supporting this idea, mitochondrial mutants with increased oxidative stress are known to show reduced pumping (Jafari et al, 2015; Yee et al, 2014).

We next investigated the host genetic pathways involved in the lifespan extension observed in worms fed FAT-7-suppressing or iron-depleted diets. Since our screen utilized FAT-7 expression, we tested whether the nuclear hormone receptor NHR-49, which regulates FAT-7 and lifespan (Naim et al, 2021; Ratnappan et al, 2014), was required for lifespan extension. The *nhr-49* loss-of-function mutant exhibited extended lifespan on FAT-7-suppressing diets, suggesting that NHR-49 is not essential for lifespan extension (Appendix Fig. S5A). Because worms experienced oxidative stress on the mutant diets (Fig. 3G,H), we asked whether oxidative stress response pathways are required for the increased lifespan. The hypoxia-inducible factor (HIF-1) is activated by ROS and is required for lifespan extension mediated by ROS (Hwang et al, 2014; Lee et al, 2010; Ravi and Singh, 2025). We examined whether HIF-1 was required for increased lifespan on the FAT-7-suppressing diets. The *hif-1* loss-of-function mutant exhibited an enhanced lifespan on the mutant diets, indicating that HIF-1 was not required for the increased lifespan (Appendix Fig. S5B).

Multiple molecular pathways regulate the response to oxidative stress. To investigate their role in extending *C. elegans* lifespan on mutant diets, we examined several key components of oxidative stress response pathways. The nuclear factor erythroid 2-related factor SKN-1 plays important roles in metabolism, aging, and orchestrating defense responses against various ROS molecules (Blackwell et al, 2015; Paek et al, 2012; Turner et al, 2024; Walker et al, 2000). We observed that *skn-1(zj15)* animals did not exhibit lifespan extension on the FAT-7-suppressing diets (Fig. 8A), indicating that SKN-1 is essential for the increased lifespan observed under these conditions. The MAP kinase kinase SEK-1, which is critical for oxidative stress response and acts upstream of SKN-1 (van der Hoeven et al, 2011; Inoue et al, 2005), was likewise

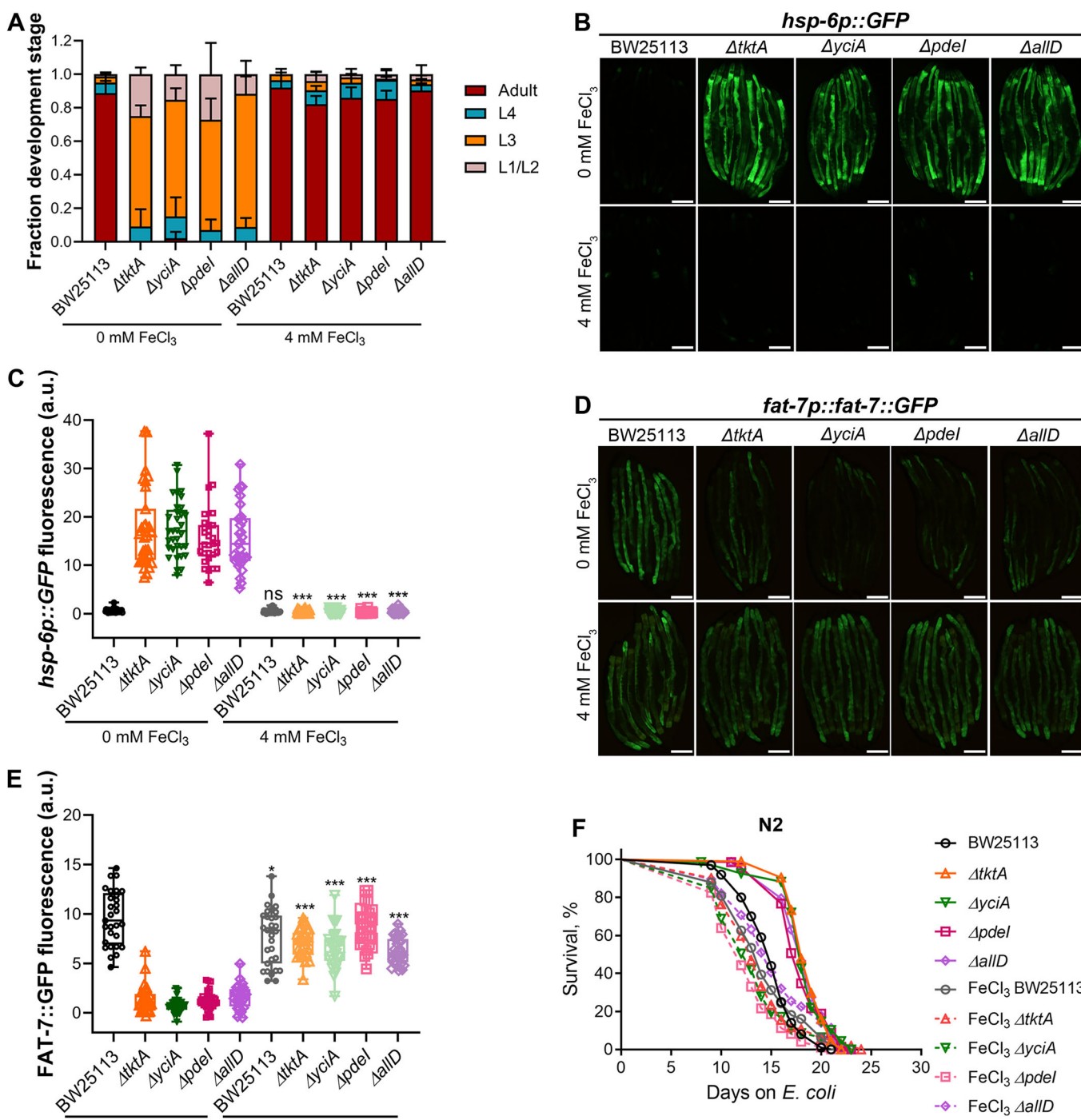

necessary for lifespan enhancement, as *sek-1(km4)* mutants did not display extended lifespan on the mutant diets (Fig. 8B). Notably, SEK-1 also regulates innate immunity and is essential for the extended lifespan observed in several long-lived *C. elegans* mutants (Soo et al, 2023). Therefore, its effect on lifespan in response to FAT-7-suppressing diets may also stem from its role in innate immune regulation. The TFEB homolog HLH-30, which is activated by multiple stresses, including oxidative stress (Lin et al, 2018), was also required for lifespan extension. The *hlh-30(tm1978)*

worms failed to show an extended lifespan on the mutant diets (Fig. 8C).

Finally, we tested whether these oxidative stress response pathways were also required for lifespan extension under iron-depleted conditions. To this end, we studied the lifespan of *skn-1(zj15)*, *sek-1(km4)*, and *hlh-30(tm1978)* animals upon iron chelation. Supplementation of bipyridyl did not increase the lifespan of *skn-1(zj15)*, *sek-1(km4)*, and *hlh-30(tm1978)* animals (Fig. 8D–F). Together, these findings demonstrated that oxidative

**Figure 6. Iron supplementation rescues mutant diet-induced phenotypes.**

(A) Quantification of different developmental stages of N2 worms grown at 20 °C for 60 h after transferring synchronized L1 larvae onto *E. coli* BW25113 and FAT-7-suppressing diets supplemented with 0 or 4 mM ferric chloride ($n = 3$ biological replicates; animals per condition per replicate >49). Data represent the mean and standard deviation from three independent experiments. (B) Representative fluorescence images of *hsp-6p::GFP* worms grown on BW25113 and FAT-7-suppressing diets supplemented with 0 or 4 mM ferric chloride. Scale bar = 200 µm. (C) Quantification of GFP levels of *hsp-6p::GFP* worms grown on BW25113 and FAT-7-suppressing diets supplemented with 0 or 4 mM ferric chloride. *P* values were calculated by comparing ferric chloride-supplemented diets (4 mM) to their respective unsupplemented controls (0 mM ferric chloride). For BW25113, $P = 0.1511$, and for all other diets, $P < 0.0001$ via the *t* test. ns nonsignificant ($n = 29$–33 worms each). (D) Representative fluorescence images of *fat-7p::fat-7::GFP* worms grown on BW25113 and FAT-7-suppressing diets supplemented with 0 or 4 mM ferric chloride. Scale bar = 200 µm. (E) Quantification of GFP levels of *fat-7p::fat-7::GFP* worms grown on BW25113 and FAT-7-suppressing diets supplemented with 0 or 4 mM ferric chloride. *P* values were calculated by comparing ferric chloride-supplemented diets (4 mM) to their respective unsupplemented controls (0 mM ferric chloride). For BW25113, $P = 0.0109$, and for all other diets, $P < 0.0001$ via the *t* test. ns nonsignificant ($n = 29$–32 worms each). (F) Representative survival curves of N2 worms fed on Δ*tktA*, Δ*yciA*, Δ*pdel*, and Δ*allD E. coli* mutants, along with the BW25113 control, supplemented with 0 or 4 mM ferric chloride. For 0 mM ferric chloride, $P < 0.001$ for Δ*tktA*, Δ*yciA*, Δ*pdel*, and Δ*allD* compared to their BW25113 control. For 4 mM ferric chloride, $P < 0.001$ for Δ*pdel*, $P < 0.05$ for Δ*allD*, and nonsignificant for Δ*tktA* and Δ*yciA* compared to their BW25113 control ($n = 3$ biological replicates; animals per condition per replicate >66). Data information: In the boxplots in (C, E), the central bands represent the median value, the boxes represent the upper and lower quartiles, and the whiskers represent the minimum and maximum values. Source data are available online for this figure.

stress response pathways, including those involving SKN-1, SEK-1, and HLH-30, are critical for lifespan extension on both mutant diets and iron-depleted conditions.

## Discussion

In this study, we identified 26 *E. coli* mutants that extend *C. elegans* lifespan. Our findings represent a distinct set of pro-longevity bacterial mutants compared to those identified in previous genome-wide screens (Appendix Fig. S1), thereby expanding our understanding of how the microbiota influence host lifespan. We found that these bacterial mutants induced oxidative stress in worms, and this elevated oxidative stress was responsible for lifespan extension. The increased oxidative stress also disrupted iron homeostasis, likely reducing the bioavailability of iron. Consistently, dietary iron limitation extended *C. elegans* lifespan (Fig. 8G), suggesting that hormetic responses are activated under these conditions.

To identify bacterial mutants that enhance *C. elegans* longevity, we used FAT-7 levels as a screening readout. The relationship between FAT-7 expression and lifespan appears to be complex. Previous studies have shown that increased FAT-7 expression correlates with lifespan extension (Han et al, 2017b). Conversely, compared to an *E. coli* diet, a *Comamonas aquatica* DA1877 diet reduces both FAT-7 expression and lifespan (Han et al, 2024; MacNeil et al, 2013). In contrast, our study shows that *E. coli* mutants that lower FAT-7 expression enhance *C. elegans* lifespan. However, our findings indicate that FAT-7 levels may not be causally linked to lifespan extension, suggesting the involvement of alternative mechanisms driving the observed longevity effects. We also showed that FAT-7 expression is suppressed by mitochondrial stress. Thus, FAT-7 suppression might have served as an indirect indicator of mitochondrial stress.

All 26 FAT-7-suppressing diets identified in our study elevated *hsp-6p::GFP* expression and extended *C. elegans* lifespan. Although UPRmt activation and lifespan extension were consistently observed across these diets, there was no strong correlation between *hsp-6p::GFP* levels and the degree of lifespan extension. The role of the UPRmt in promoting longevity remains controversial (Bennett et al, 2014; Soo et al, 2021; Wu et al, 2018). For instance, gain-of-function mutations in *atfs-1* have been shown to reduce lifespan (Bennett et al, 2014; Soo et al, 2021). However, a recent study demonstrated that mild UPRmt activation can extend

lifespan, whereas strong activation has the opposite effect (Di Pede et al, 2025). These findings suggest that UPRmt contributes to longevity only under specific conditions and at specific activation levels. In our study, lifespan extension on FAT-7-suppressing diets was dependent on ATFS-1, indicating that UPRmt activation was necessary for this effect.

A previous screen by Zhang et al identified 244 *E. coli* mutants that delayed *C. elegans* development, with most of these mutants also increasing *hsp-6* expression (Zhang et al, 2019). The *E. coli* mutants identified in our screen exhibited the same phenotypes in *C. elegans* and showed a nearly complete overlap with those identified by Zhang et al However, our screen identified far fewer mutants. One possible explanation is that the *fat-7::GFP* strain used in our study has low baseline GFP fluorescence on the control diet, potentially leading to the exclusion of mutant diets that caused only mild reductions in FAT-7 levels. Zhang et al reported that their identified *E. coli* mutants exhibited high ROS levels, which could lead to iron depletion in *C. elegans* (Zhang et al, 2019). We found that *C. elegans* fed on our *E. coli* mutants also exhibited elevated ROS levels. Iron supplementation restored all mutant diet-induced phenotypes to control levels, suggesting that iron limitation may underlie these effects. Consistently, iron chelation in control diets recapitulated the same phenotypes observed with mutant diets.

Iron is an essential trace element required for various cellular processes, including oxygen transport, energy metabolism, DNA synthesis, and gene regulation (Wang and Pantopoulos, 2011). Maintaining optimal iron levels is crucial for cellular homeostasis, as both iron deficiency and excess can be detrimental (Galaris et al, 2019; Zhang et al, 2019). Interestingly, both high and low-iron levels have been shown to extend *C. elegans* lifespan, possibly through hormetic responses that activate stress-related pathways (Anand et al, 2020; Bhat et al, 2024; Schiavi et al, 2015). One potential mechanism by which iron depletion extends lifespan is through reduced ferroptosis, a form of iron-dependent cell death (Jenkins et al, 2020; Kim et al, 2022). Alternatively, iron depletion may disrupt iron-sulfur cluster formation, a process that has been linked to lifespan extension in *C. elegans* (Sheng et al, 2021).

Frataxin, a key protein involved in iron-sulfur cluster biogenesis, has been implicated in lifespan regulation (Ast et al, 2019; Schiavi et al, 2013). Inhibition of frataxin extends *C. elegans* lifespan, potentially by disrupting iron-sulfur cluster formation (Schiavi et al, 2023, 2015). Frataxin silencing promotes longevity through

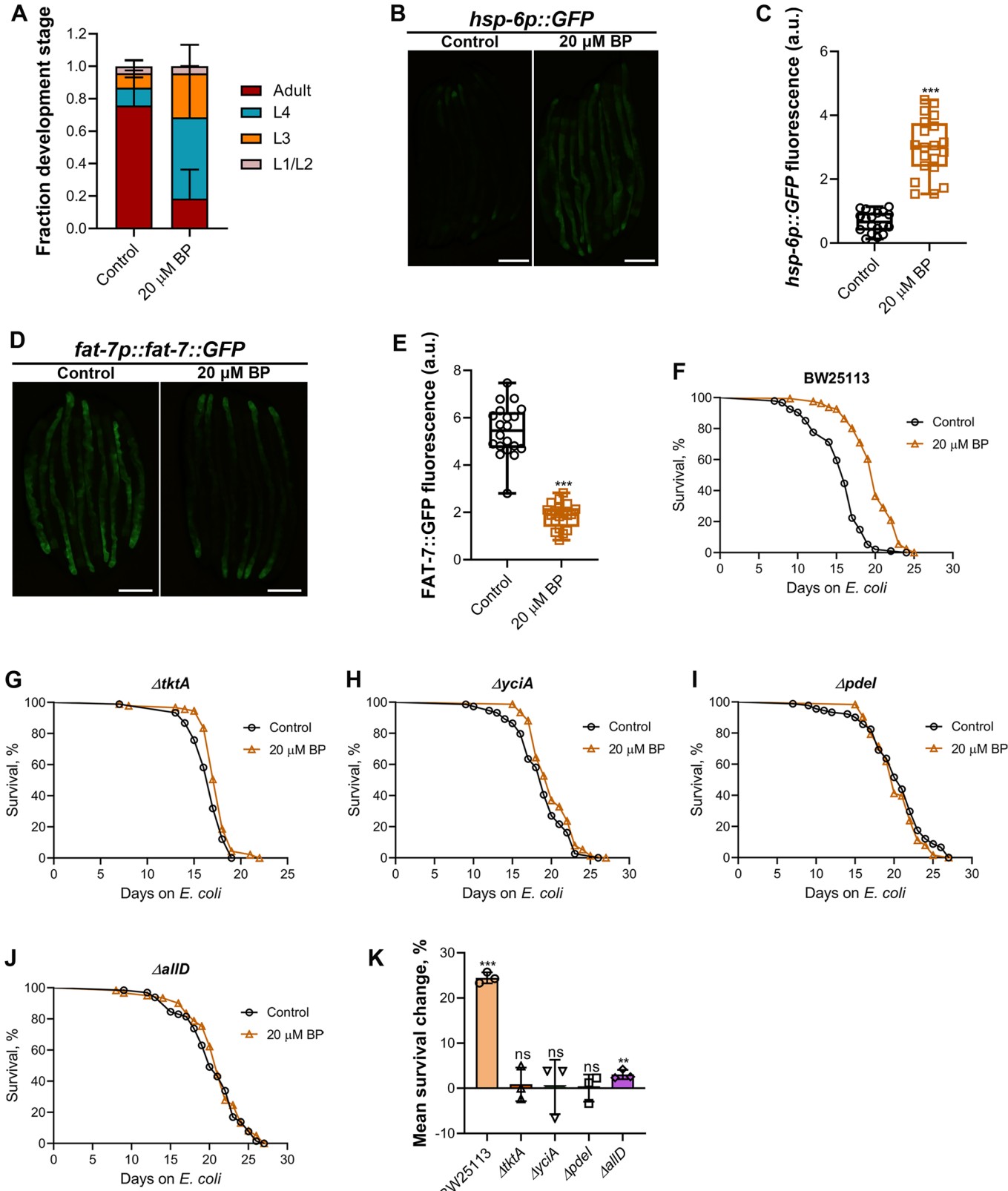

multiple mechanisms, including mitophagy activation, HIF-1 signaling, and ferroptosis inhibition (Schiavi et al, 2023, 2015, 2013). While iron depletion appears to mediate some of the effects of frataxin inhibition, the two processes also involve

distinct mechanisms. For instance, frataxin inhibition extends lifespan via HIF-1 activation, whereas iron chelation does so independently of HIF-1 (Schiavi et al, 2015). Similarly, the bacterial mutants identified in our study enhanced lifespan in a HIF-1-

**Figure 7.  Low dietary iron mimics mutant diet-induced phenotypes.**

(A) Quantification of different developmental stages of N2 worms grown at 20 °C for 60 h after transferring synchronized L1 larvae onto *E. coli* BW25113 supplemented with DMSO control and 20 μM 2,2'-bipyridyl (BP) ($n = 3$ biological replicates; animals per condition per replicate >60). Data represent the mean and standard deviation from three independent experiments. (B) Representative fluorescence images of *hsp-6p::GFP* worms grown on *E. coli* BW25113 supplemented with DMSO control and 20 μM BP. Scale bar =  200 μm. (C) Quantification of GFP levels of *hsp-6p::GFP* worms grown on *E. coli* BW25113 supplemented with DMSO control and 20 μM BP. ***$P < 0.0001$ via the *t* test ($n = 18$–20 worms each). (D) Representative fluorescence images of *fat-7p::fat-7::GFP* worms grown on *E. coli* BW25113 supplemented with DMSO control and 20 μM BP. Scale bar =  200 μm. (E) Quantification of GFP levels of *fat-7p::fat-7::GFP* worms grown on *E. coli* BW25113 supplemented with DMSO control and 20 μM BP. ***$P < 0.0001$ via the *t* test ($n = 20$ worms each). (F–J) Representative survival curves of N2 worms grown on BW25113 (F), *ΔtktA* (G), *ΔyciA* (H), *Δpdel* (I), and *ΔallD* (J) mutant *E. coli* diets supplemented with DMSO control and 20 μM BP. $P < 0.001$ for BW25113, $P < 0.001$ for *ΔtktA* and nonsignificant for *ΔyciA*, *Δpdel*, and *ΔallD* ($n = 3$ biological replicates; animals per condition per replicate >50). (K) The percent change in mean survival of N2 worms grown on BW25113, *ΔtktA*, *ΔyciA*, *Δpdel*, and *ΔallD* mutant *E. coli* diets supplemented with 20 μM BP compared to their respective DMSO controls. The *P* values compared to their respective controls are the following: BW25113 < 0.0001, *tktA* = 0.7051, *yciA* = 0.9360, *pdel* = 0.9792, and *allD* = 0.0069 via the *t* test. ns, nonsignificant. Data represent the mean and standard deviation from three independent experiments. Data information: In the boxplots in (C, E), the central bands represent the median value, the boxes represent the upper and lower quartiles, and the whiskers represent the minimum and maximum values. Source data are available online for this figure.

independent manner, suggesting that frataxin inhibition may activate additional pathways beyond iron limitation.

We observed that *C. elegans* fed on mutant diets exhibited elevated ROS levels. While increased ROS can promote *C. elegans* longevity (Hwang et al, 2014; Lee et al, 2010; Schulz et al, 2007; Yang and Hekimi, 2010), its effects on lifespan are complex and context-dependent, involving multiple pathways (Hwang et al, 2014; Schaar et al, 2015; Yang and Hekimi, 2010). For example, depending on the type of ROS involved, HIF-1 may or may not be required for ROS-mediated lifespan extension (Lee et al, 2010; Yang and Hekimi, 2010). Likewise, the role of SKN-1 in ROS-mediated longevity varies depending on the specific context (Hwang et al, 2014; Wei and Kenyon, 2016; Yang and Hekimi, 2010). We found that the oxidative stress response pathways SKN-1, SEK-1, and HLH-30 were essential for lifespan extension on mutant diets and under iron-depleted conditions. SEK-1, a MAPK kinase, is an upstream regulator of SKN-1 (van der Hoeven et al, 2011; Inoue et al, 2005), suggesting that these two factors may act in the same pathway to regulate lifespan under iron-limited conditions. Although HLH-30 is known to be activated by oxidative stress (Lin et al, 2018), its role in ROS-mediated lifespan extension had not been previously investigated. Our findings suggest that HLH-30 plays a key role in lifespan extension under both high ROS and low-iron conditions. It is also possible that additional factors from the *E. coli* mutant diets identified in our study contribute to *C. elegans* lifespan extension. Future research investigating bacterial metabolites from these *E. coli* mutants could provide further insights into how gut microbiota influences host longevity.

# Methods

### Reagents and tools table

| Reagent/resource | Reference or source | Identifier or catalog number |
|---|---|---|
| **Experimental models** | | |
| *Escherichia coli* OP50 | *Caenorhabditis* Genetics Center (CGC) | OP50 |
| *E. coli* HT115(DE3) containing L4440 plasmid | Source BioScience | RNAi Empty Vector |
| *E. coli* HT115(DE3) containing *tomm-22* RNAi plasmid | Ahringer RNAi library | *tomm-22* RNAi |

| Reagent/resource | Reference or source | Identifier or catalog number |
|---|---|---|
| *E. coli* BW25113 | Rachna Chaba laboratory | BW25113 |
| *E. coli* Keio collection | Horizon Discovery | Keio collection |
| *C. elegans*: N2 | CGC | N2 |
| *C. elegans*: *nIs590 [fat-7p::fat-7::GFP + lin15(+)]* | CGC | DMS303 |
| *C. elegans*: *fat-2(wa17)* | CGC | BX26 |
| *C. elegans*: *zcIs13 [hsp-6p::GFP + lin-15(+)]* | CGC | SJ4100 |
| *C. elegans*: *atfs-1(gk3094)* | CGC | VC3201 |
| *C. elegans*: *skn-1(zj15)* | CGC | QV225 |
| *C. elegans*: *sek-1(km4)* | CGC | KU4 |
| *C. elegans*: *hlh-30(tm1978)* | CGC | JIN1375 |
| *C. elegans*: *nhr-49(nr2041)* | CGC | STE68 |
| *C. elegans*: *isp-1(qm150)* | CGC | MQ887 |
| *C. elegans*: *hif-1(ia4)* | CGC | ZG31 |
| *C. elegans*: *eat-2(ad465)* | CGC | DA465 |
| *C. elegans*: *eat-2(ad465); nIs590 [fat-7p::fat-7::GFP + lin-15(+)]* | This study | "Methods" |
| *C. elegans*: *eat-2(ad465); zcIs13 [hsp-6p::GFP + lin-15(+)]* | This study | "Methods" |
| *C. elegans*: *jsnEx4 [vha-6p::fat-7+myo-3p::mCherry]* | This study | "Methods" |
| **Recombinant DNA** | | |
| Plasmid- pCFJ104 (Pmyo-3::mCherry::unc-54) | Addgene | pCFJ104 |
| Plasmid- pPD95_77 (Empty backbone) | Addgene | pPD95_77 |
| Plasmid- *vha-6p::fat-7* | This study | "Methods" |
| **Antibodies** | | |
| **Oligonucleotides and other sequence-based reagents** | | |
| fat-7_cDNA_cloning_F | CGCGTCGACAAATGACGGTAAAAACTCGC | |
| fat-7_cDNA_cloning_R | GCGGGTACCCTTTTATGGACAACCAACGC | |
| **Chemicals, enzymes, and other reagents** | | |
| 2,2'-Bipyridyl | HiMedia | Cat# GRM791 |
| 2,7'-Dichlorofluorescein diacetate | Sigma | Cat# 35845 |
| 5-Fluoro-2'-deoxyuridine | Thermo Fisher Scientific | Cat# ALF-L16497-ME |

| Reagent/resource | Reference or source | Identifier or catalog number |
|---|---|---|
| Ampicillin sodium salt | HiMedia | Cat# TC021 |
| DMSO | HiMedia | Cat# TC185 |
| Ferric chloride | HiMedia | Cat# TC583 |
| Isopropyl-b-D-thiogalactopyranoside | HiMedia | Cat# MB072 |
| Kanamycin sulphate | HiMedia | Cat# MB105 |
| N-acetyl-cysteine | HiMedia | Cat# RM3142 |
| Nonidet P-40 | HiMedia | Cat# MB143 |
| Paraquat dichloride | Sigma | Cat# 856177 |
| Sodium oleate | Sigma | Cat# O7501 |
| HindIII | Takara | Cat# 1615 |
| KpnI | Takara | Cat# 1618 |
| SalI | Takara | Cat# 1636 |
| **Software** | | |
| GraphPad Prism 8 | GraphPad Software | RRID: SCR_002798 |
| Photoshop CS5 | Adobe | RRID: SCR_014199 |
| ImageJ | NIH | RRID: SCR_003070 |
| BioRender | https://www.biorender.com/ | |
| BioVenn | https://www.biovenn.nl/ Hulsen et al, 2008 | |
| InteractiVenn | https://www.interactivenn.net/ Heberle et al, 2015 | |
| Hypergeometric *P* value calculator | https://systems.crump.ucla.edu/hypergeometric/ | |
| **Other** | | |

## Bacterial strains

The bacterial strains used in this study include *Escherichia coli* OP50, *E. coli* HT115(DE3), *E. coli* BW25113, and mutants from the *E. coli* Keio collection (Baba et al, 2006). *E. coli* OP50 and *E. coli* BW25113 cultures were grown in Luria-Bertani (LB) broth at 37 °C. The Keio collection mutants were grown in LB broth supplemented with 25 µg/mL kanamycin at 37 °C during the FAT-7::GFP screen, while for other experiments, they were grown in LB without the antibiotics.

## *C. elegans* strains and growth conditions

*C. elegans* hermaphrodites were maintained on NGM plates seeded with *E. coli* OP50 at 20 °C unless otherwise specified. The Bristol N2 strain was used as the wild-type control unless indicated otherwise. The following strains were used in this study: DMS303 nIs590 [*fat-7p::fat-7::GFP + lin15(+)*], BX26 *fat-2(wa17)*, SJ4100 zcIs13 [*hsp-6p::GFP + lin-15( + )*], VC3201 *atfs-1(gk3094)*, QV225 *skn-1(zj15)*, KU4 *sek-1(km4)*, JIN1375 *hlh-30(tm1978)*, STE68 *nhr-49(nr2041)*, ZG31 *hif-1(ia4)*, MQ887 *isp-1(qm150)*, *eat-2(ad465);nIs590* [*fat-7p::fat-7::GFP + lin15(+)*], and *eat-2(ad465);zcIs13* [*hsp-6p::GFP + lin-15(+)*]. Some of the strains were obtained from the Caenorhabditis Genetics Center (University of Minnesota, Minneapolis, MN). The *eat-2(ad465);nIs590* [*fat-7p::fat-7::GFP + lin15(+)*] and *eat-2(ad465);zcIs13* [*hsp-6p::GFP + lin-15(+)*] strains were generated using standard genetic crosses.

For all the experiments, worms were synchronized by bleach treatment to obtain the same-stage L1 larvae.

## Plasmid constructs and generation of transgenic *C. elegans*

For the overexpression of *fat-7*, the *fat-7* gene was amplified using the cDNA of N2 worms. The gene, including its stop codon, was cloned into the pPD95_77 plasmid using the restriction sites SalI and KpnI. The promoter region of the intestine-specific gene *vha-6* (1248 bp upstream) was cloned upstream of *fat-7* using the restriction sites HindIII and SalI. N2 worms were microinjected with *vha-6p::fat-7* plasmid along with pCFJ104 (*myo-3p::mCherry*) as a coinjection marker to generate the overexpression strain, *jsnEx4* [*vha-6p::fat-7 + myo-3p::mCherry*]. The *vha-6p::fat-7* plasmid was used at a concentration of 50 ng/µL, while the coinjection marker was used at 25 ng/µL.

## Supplementation experiments

The following supplements were obtained from HiMedia BioSciences: ferric chloride (#TC583), N-acetylcysteine (NAC) (#RM3142), and 2,2'-bipyridyl (#GRM791). Paraquat dichloride (PQ) (#856177) and sodium oleate (# O7501) were purchased from Sigma. Stock solutions were prepared as follows: 1 M ferric chloride, 0.5 M PQ, and 0.5 M NAC in water, and 100 mM 2,2'-bipyridyl in dimethyl sulfoxide (DMSO). All stock solutions were stored at −20 °C and diluted to their final concentrations in NGM before pouring the plates. For sodium oleate supplementation, Nonidet P-40 was added to a final concentration of 0.001% in liquid NGM before autoclaving, in both supplemented and control plates. Sodium oleate was weighed and added directly to the NGM before pouring plates. For experiments with 2,2'-bipyridyl, control plates were supplemented with an equivalent amount of DMSO. Worms were grown from the synchronized L1 stage on all supplements except PQ. For PQ treatment, worms were exposed at the late L4 stage and incubated for 24 h prior to fluorescence imaging.

## *E. coli* deletion mutant screening for diets that modulate *C. elegans* FAT-7 levels

Bacterial mutants were grown overnight at 37 °C in LB broth supplemented with 25 µg/mL kanamycin in 96-well plates. Subsequently, 30 µL of the overnight cultures were seeded onto 24-well plates containing NGM agar supplemented with 25 µg/mL kanamycin. The plates were incubated at room temperature for at least 2 days to allow bacterial growth before use in experiments. For screening, embryos of the *fat-7p::fat-7::GFP* strain were harvested from gravid adults using an alkaline bleach solution and incubated in M9 buffer at room temperature for 22 h to obtain synchronized L1 larvae. Approximately 30–40 synchronized L1 larvae were transferred to each well of 24-well NGM agar plates seeded with individual *E. coli* single-gene deletion mutants and incubated at 20 °C. Since FAT-7 levels vary across developmental stages, screening was performed at the day-1-adult stage to identify mutants that modulate FAT-7 expression. GFP fluorescence was monitored in each well to identify *E. coli* mutants that either enhanced or suppressed FAT-7 levels compared to the *E. coli* BW25113 control diet.

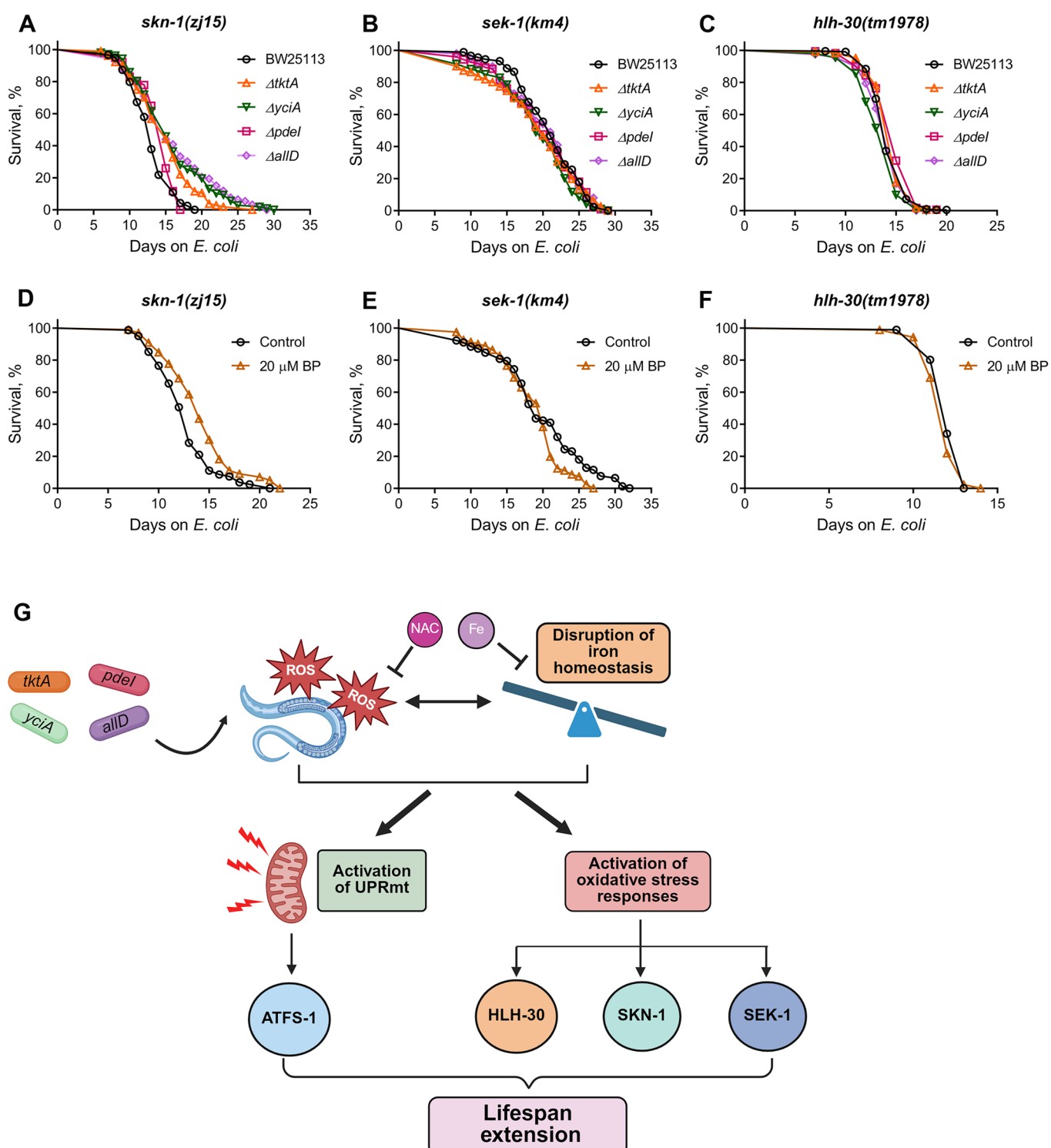

Bacterial mutants identified as hits in the initial screen were retested three times on individual NGM plates seeded with the corresponding mutants. *E. coli* mutants that consistently reproduced the phenotype across three independent trials were considered as primary hits. Gene ontology analysis was performed using the DAVID Bioinformatics Database (https://david.ncifcrf.gov/tools.jsp).

## Bacterial growth curve assay

Primary cultures of each bacterial strain were grown in LB broth without antibiotics at 37 °C with shaking for 12 h. These cultures were then diluted to an initial optical density ($OD_{600}$) of 0.01 in 15 mL of fresh LB broth in 50-mL centrifuge tubes. The diluted cultures were incubated at 37 °C with continuous shaking, and

**Figure 8. Lifespan extension under low dietary iron depends on oxidative stress response pathways.**

(A) Representative survival curves of *skn-1(zj15)* worms fed on *ΔtktA, ΔyciA, Δpdel*, and *ΔallD E. coli* mutants along with the BW25113 control. *P* < 0.001 for *ΔtktA, ΔyciA*, and *ΔallD* and *P* < 0.05 for *Δpdel* compared to the BW25113 control (*n* = 3 biological replicates; animals per condition per replicate >70). (B) Representative survival curves of *sek-1(km4)* worms fed on *ΔtktA, ΔyciA, Δpdel*, and *ΔallD E. coli* mutants along with the BW25113 control. *P* < 0.05 for *ΔyciA* and nonsignificant for *ΔtktA, Δpdel*, and *ΔallD* compared to the BW25113 control (*n* = 3 biological replicates; animals per condition per replicate >60). (C) Representative survival curves of *hlh-30(tm1978)* worms fed on *ΔtktA, ΔyciA, Δpdel*, and *ΔallD E. coli* mutants along with the BW25113 control. *P* < 0.001 for *ΔyciA*, *P* < 0.05 for *ΔtktA* and *Δpdel*, and nonsignificant for *ΔallD* compared to the BW25113 control (*n* = 3 biological replicates; animals per condition per replicate >80). (D) Representative survival curves of *skn-1(zj15)* worms grown on *E. coli* BW25113 supplemented with DMSO control and 20 μM 2,2'-bipyridyl (BP). *P* < 0.001 for BP compared to the control (*n* = 3 biological replicates; animals per condition per replicate >60). (E) Representative survival curves of *sek-1(km4)* worms grown on *E. coli* BW25113 supplemented with DMSO control and 20 μM BP. nonsignificant for BP compared to the control (*n* = 3 biological replicates; animals per condition per replicate >70). (F) Representative survival curves of *hlh-30(tm1978)* worms grown on *E. coli* BW25113 supplemented with DMSO control and 20 μM BP. nonsignificant for BP compared to the control (*n* = 3 biological replicates; animals per condition per replicate >80). (G) Model showing the mechanism of *C. elegans* lifespan extension by *E. coli* mutants. The model was created using BioRender. Source data are available online for this figure.

$OD_{600}$ measurements were taken hourly to monitor bacterial growth.

## RNA interference

RNAi was performed to generate loss-of-function phenotypes by feeding nematodes the *E. coli* strain HT115(DE3) expressing double-stranded RNA homologous to *tomm-22*. RNAi was carried out as described previously (Das et al, 2024; Rao et al, 2025). Briefly, *E. coli* strains were cultured overnight in LB containing ampicillin (100 μg/mL) at 37 °C. They were then concentrated 20 times and plated on an NGM plate containing 3 mM isopropyl β-D-thiogalactoside and ampicillin (100 μg/mL) (RNAi plate). The plated bacteria were allowed to grow overnight at 37 °C before use. For worm synchronization, gravid adults were bleached, and embryos were allowed to hatch in M9 buffer for 22 h at room temperature to obtain L1 larvae. These synchronized L1s were transferred to RNAi plates and incubated at 20 °C till the day-1-adult stage. The *tomm-22* RNAi clone was obtained from the Ahringer RNAi library.

## *C. elegans* longevity assays

Lifespan assays were conducted as described earlier (Das et al, 2024). Briefly, gravid adults were lysed using an alkaline bleach solution to obtain embryos, which were then incubated in M9 buffer for 20–24 h to synchronize them at the L1 larval stage. Synchronized L1 larvae were transferred to NGM plates seeded with either wild-type *E. coli* BW25113 or bacterial mutants identified from the FAT-7::GFP screen. For assays involving ferric chloride, NAC, sodium oleate, and 2,2'-bipyridyl supplementation, synchronized L1 larvae were transferred to NGM plates containing these supplements. At the late L4 larval stage, the animals were transferred to corresponding bacterial diet plates or supplement plates containing 50 μg/mL FUdR and incubated at 20 °C. Worms were monitored daily or every other day and scored as alive or dead. Animals that failed to exhibit touch-provoked movement were classified as dead, while those that crawled off the plates were censored from the analysis. For the lifespan analysis, young adult animals were designated as day 0. Three independent experiments were performed for each condition.

## *C. elegans* development assays

Gravid N2 hermaphrodites were lysed with an alkaline bleach solution to isolate eggs, which were then incubated in M9 buffer at

room temperature for 22 h. Approximately 50–100 synchronized L1 larvae were transferred onto NGM plates seeded with either the *E. coli* BW25113 control or mutant diets and incubated at 20 °C for 60 h. The assays were similarly carried out for NAC and ferric chloride-supplemented diets. Animals at various developmental stages (L1/L2, L3, L4, and adult) were subsequently quantified. The experiment was repeated in at least three independent biological replicates.

## Pharyngeal pumping assay

Pharyngeal pumping rates were measured in 1-day-old adult animals grown on *E. coli* BW25113 or the FAT-7-suppressing diets. The number of terminal bulb contractions was counted over a 30-s interval for each worm. For each condition, ten worms were assayed, and the experiment was performed in three independent biological replicates.

## Quantification of reactive oxygen species (ROS) levels

ROS levels were quantified using 2',7'-dichlorofluorescein diacetate (DCFHDA, Sigma-Aldrich #35845). A 50 mM DCFHDA stock was prepared in DMSO and stored at −20 °C. Before each experiment, a 50 μM DCFHDA working solution was freshly prepared in M9 buffer. Synchronized L1 larvae of N2 worms were obtained as described above and grown on the *E. coli* BW25113 control and mutant diets until the day-1 adult stage at 20 °C. Subsequently, 15–20 worms were transferred to 150 μL M9 buffer, followed by the addition of 150 μL of the DCFHDA working solution, resulting in a final DCFHDA concentration of 25 μM. Samples were incubated in the dark at room temperature for 5 h with gentle shaking. Next, the worms were pelleted, the supernatant was removed, and the worms were washed twice with PBS containing 0.01% Triton X-100. The prepared samples were then subjected to fluorescence imaging. The 2',7'-dichlorofluorescein (DCF) fluorescence was visualized using a GFP filter on a fluorescence microscope. At least five worms per condition were imaged, and three independent biological replicates were performed.

## Fluorescence imaging

Fluorescence imaging was carried out as described previously (Gokul and Singh, 2022; Ravi et al, 2023). Briefly, animals were anesthetized using M9 buffer containing 50 mM sodium azide and placed on 2% agarose pads. The animals were then visualized using either a Nikon SMZ-1000 or SMZ18 fluorescence stereomicroscope.

## RNA sequencing and data analysis

Synchronized L1 larvae were obtained from wild-type animals as described above and grown on the *E. coli* BW25113 control diet and four mutant diets, including Δ*tktA*, Δ*pdeI*, Δ*yciA*, and Δ*allD*, until the day 1 adult stage. Total RNA was extracted from three biological replicates using the RNeasy Plus Universal Kit (Qiagen, the Netherlands). Library preparation and sequencing were performed at Unipath Specialty Laboratory Ltd., India. cDNA libraries were sequenced on the NovaSeq 6000 platform using 150-bp paired-end reads.

RNA sequencing data were processed and analyzed using the Galaxy web platform (https://usegalaxy.org/), as described previously (Ghosh and Singh, 2024; Rao et al, 2025). Paired-end reads were first trimmed using the Trimmomatic tool and aligned to the *C. elegans* genome (WS220) with the STAR aligner. Gene expression levels were quantified using htseq-count, and differential expression analysis was performed using DESeq2. Genes with at least a twofold change and $P < 0.01$ were considered differentially expressed. Gene ontology enrichment analysis was conducted using the DAVID Bioinformatics Database. Venn diagrams were generated using the tools InteractiVenn (Heberle et al, 2015) and BioVenn (Hulsen et al, 2008). The enrichment factor and $P$ values for overlap were generated using the hypergeometric $P$ value calculator (https://systems.crump.ucla.edu/hypergeometric/). For the calculation, the total number of genes was set to 20,000.

## Quantification and statistical analysis

Statistical analyses were performed with Prism 8 (GraphPad). All error bars represent the mean ± standard deviation (SD). An unpaired, two-tailed, two-sample *t* test was used when applicable, with statistical significance set at $P < 0.05$. In the figures, statistical significance is indicated by asterisks: $*P < 0.05$, $**P < 0.01$, and $***P < 0.001$, relative to the relevant controls. Survival fractions were calculated using the Kaplan–Meier method, and statistical significance between survival curves was determined using the log-rank test. For oleic acid supplementation experiments across different mutant diets, a multivariable Cox regression analysis was performed with genotype and oleic acid supplementation as covariates, and corresponding hazard ratios and $P$ values were calculated. All experiments were performed in triplicate unless indicated otherwise.

## Data availability

The RNA sequencing data for N2 worms grown on the wild-type, Δ*tktA*, Δ*pdeI*, Δ*yciA*, and Δ*allD* *E. coli* BW25113 have been submitted to the public repository, the Sequence Read Archive, with BioProject ID PRJNA1219049. All data generated or analyzed during this study are included in the manuscript and supporting files.

The source data of this paper are collected in the following database record: biostudies:S-SCDT-10_1038-S44318-025-00634-7.

## Peer review information

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

## Acknowledgements

We thank the Caenorhabditis Genetics Center (funded by the NIH Office of Research Infrastructure Programs (P40 OD010440)) for providing the strains used in this study. We thank Rajneesh Rao for assistance with some lifespan assays and Annesha Ghosh for cloning the *vha-6* promoter. This work was supported by the following grants: Anusandhan National Research Foundation (ANRF) Core Research Grant (Ref. No. CRG/2023/001136) awarded by DST, India; Har-Gobind Khorana-Innovative Young Biotechnologist Fellowship (File No. HRD-17011/2/2023-HRD-DBT) and Ramalingaswami Re-entry Fellowship (Ref. No. BT/RLF/Re-entry/50/2020) awarded by the Department of

Biotechnology, India; STARS grant (File No. MoE-STARS/STARS-2/2023-0116) awarded by the Ministry of Education, India; Research Grant (Ref. No. 37/1741/23/EMR-II) awarded by the Council of Scientific & Industrial Research (CSIR), India; and IISER Mohali intramural funds. PD was supported by a senior research fellowship from the CSIR, India.

## Author contributions

**Priyanka Das**: Conceptualization; Data curation; Formal analysis; Investigation; Visualization; Methodology; Writing—original draft; Writing—review and editing. **Ravi**: Validation; Investigation. **Jogender Singh**: Conceptualization; Data curation; Formal analysis; Supervision; Funding acquisition; Visualization; Writing—original draft; Writing—review and editing.

Source data underlying figure panels in this paper may have individual authorship assigned. Where available, figure panel/source data authorship is listed in the following database record: biostudies:S-SCDT-10_1038-S44318-025-00634-7.

## Disclosure and competing interests statement

The authors declare no competing interests.

# Expanded View Figures

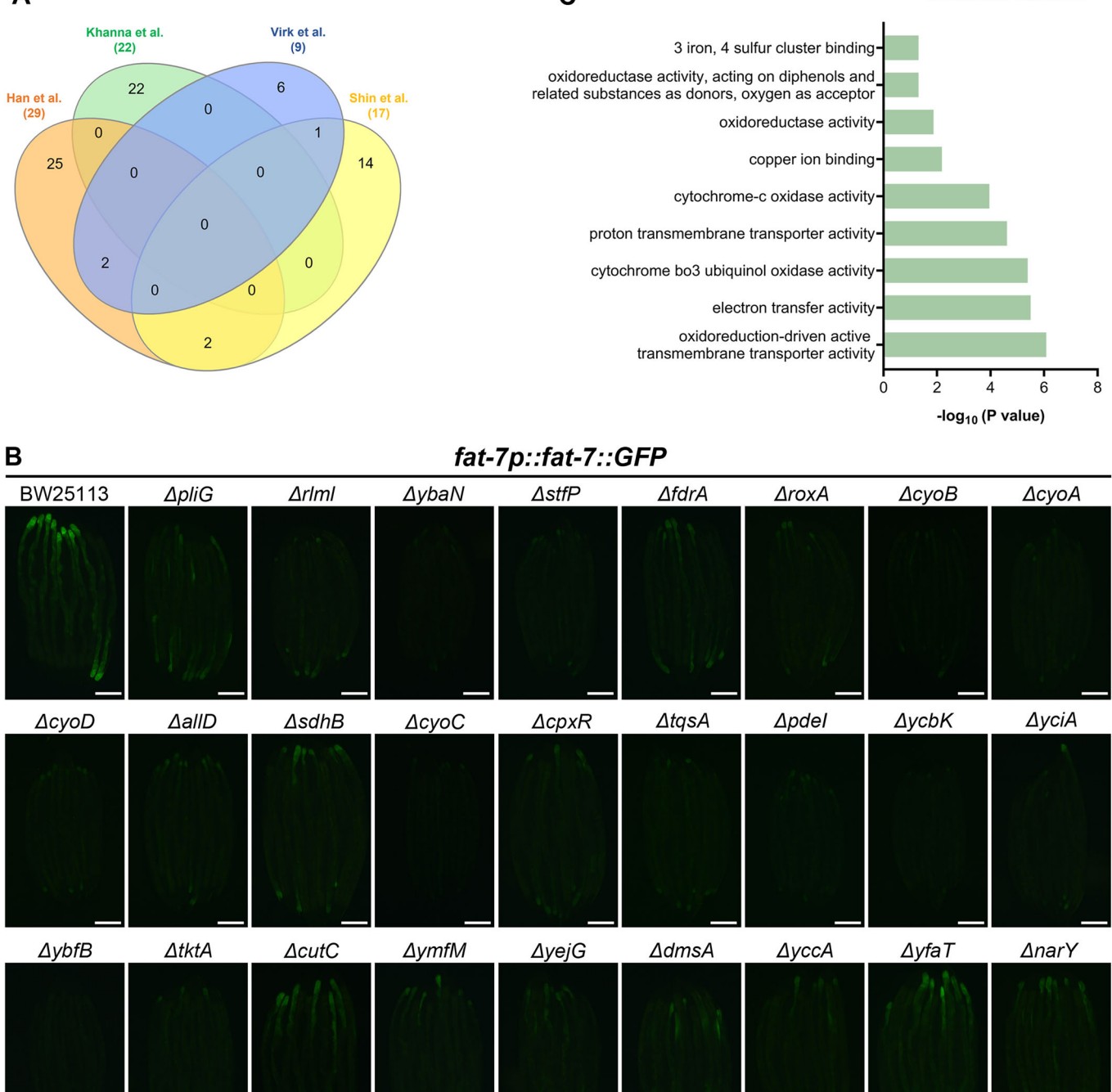

**Figure EV1. Genome-wide bacterial screen identifies *E. coli* mutants that modulate *C. elegans* FAT-7 levels.**

(A) Venn diagram showing the overlap among *E. coli* mutants identified in genome-wide bacterial screens for mutants that extend *C. elegans* lifespan (Han et al, 2017a; Khanna et al, 2016; Shin et al, 2020; Virk et al, 2016). (B) Representative fluorescence images of *fat-7p::fat-7::GFP* worms grown on *E. coli* BW25113 and mutant diets. Scale bar = 200 μm. (C) Gene ontology enrichment analysis for molecular functions associated with the 26 *E. coli* mutants that downregulate *C. elegans* FAT-7 levels. The statistical analysis was performed using Fisher's exact test. Source data are available online for this figure.

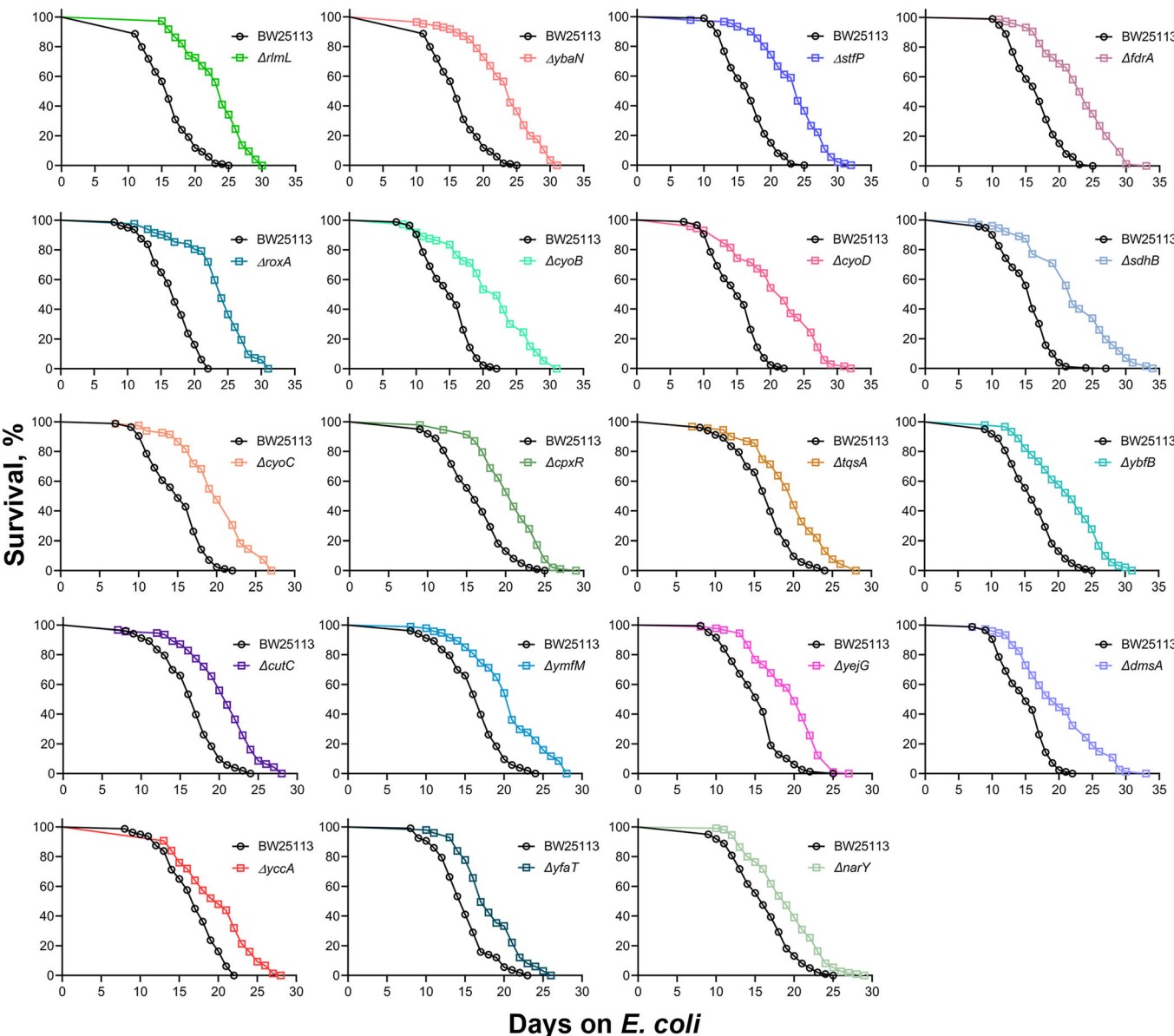

**Figure EV2.** *E. coli* **mutants that decrease FAT-7 levels extend** *C. elegans* **lifespan.**

Representative survival curves of N2 worms fed on *E. coli* BW25113 and mutants that decrease FAT-7 levels. The BW25113 control is common for different cohorts of the survival curves. The survival curves for the remaining mutants are shown in Figs. 2A–C and 3A. Source data are available online for this figure.

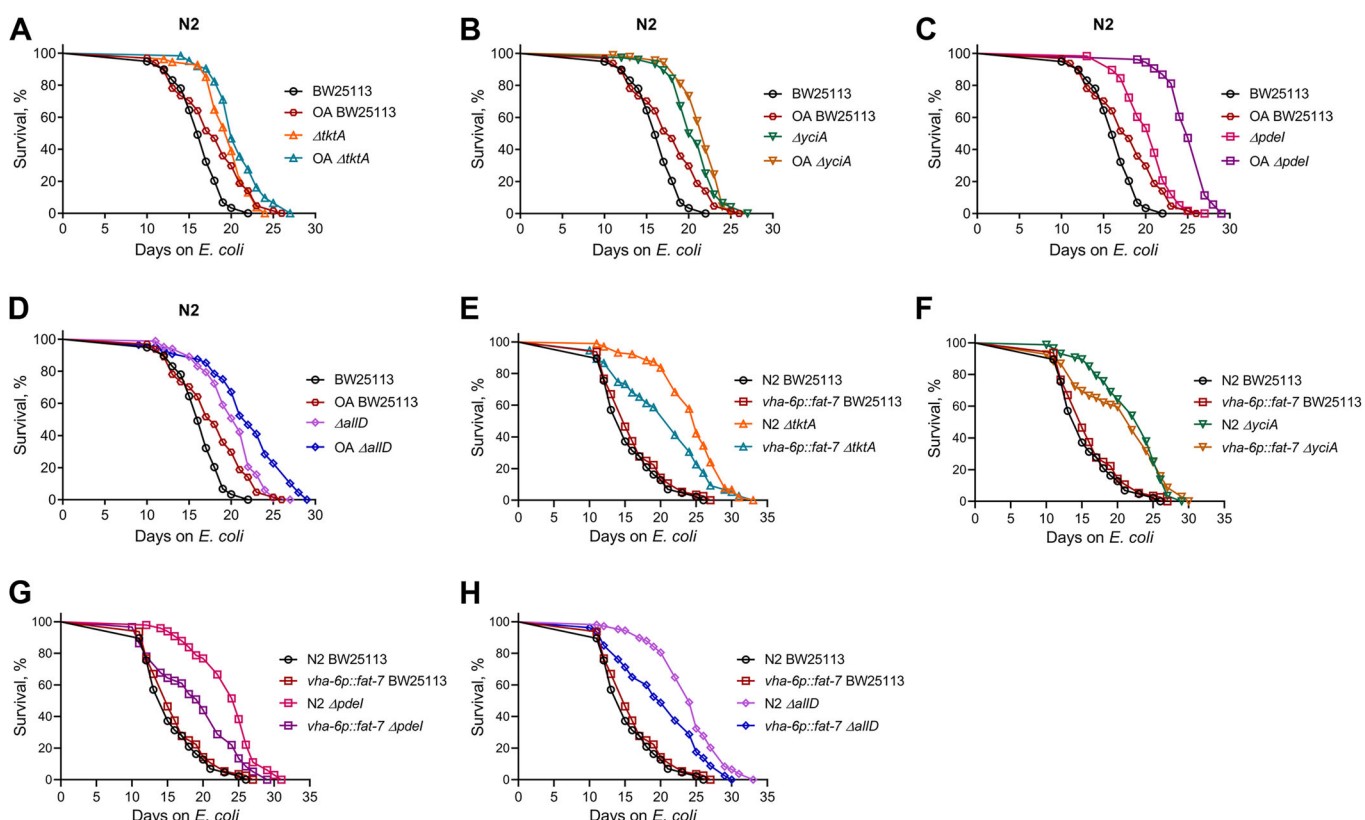

**Figure EV3. *E. coli* mutants do not extend *C. elegans* lifespan via oleic acid.**

(A–D) Representative survival curves of N2 worms fed on *E. coli* BW25113 with or without 2 mM oleic acid (OA), along with N2 worms fed on *ΔtktA* (A), *ΔyciA* (B), *Δpdel* (C), and *ΔallD* (D) *E. coli* mutants with or without 2 mM OA. The BW25113 control is common for all the panels. $P < 0.001$ for *ΔtktA, ΔyciA, Δpdel*, and *ΔallD* compared to their respective control BW25113 ($n = 3$ biological replicates; animals per condition per replicate >52). The hazard ratios and $P$ values calculated by Cox regression analysis on each of the mutant diets for their interaction with oleic acid supplementation are the following: *tktA*:OA (hazard ratio = 1.629, $P = 0.0669$), *yciA*:OA (hazard ratio = 2.356, $P = 0.0006$), *pdel*:OA (hazard ratio = 0.4756, $P = 0.0094$) and *allD*:OA (hazard ratio = 1.096, $P = 0.711$). (E–H) Representative survival curves of N2 and *vha-6p::fat-7* worms grown on *E. coli* BW25113, along with N2 and *vha-6p::fat-7* worms grown on *ΔtktA* (E), *ΔyciA* (F), *Δpdel* (G), and *ΔallD* (H) *E. coli* mutants. The BW25113 control is common for all the panels. $P < 0.001$ for *ΔtktA, ΔyciA, Δpdel*, and *ΔallD* compared to their respective control BW25113 ($n = 3$ biological replicates; animals per condition per replicate >49). Source data are available online for this figure.

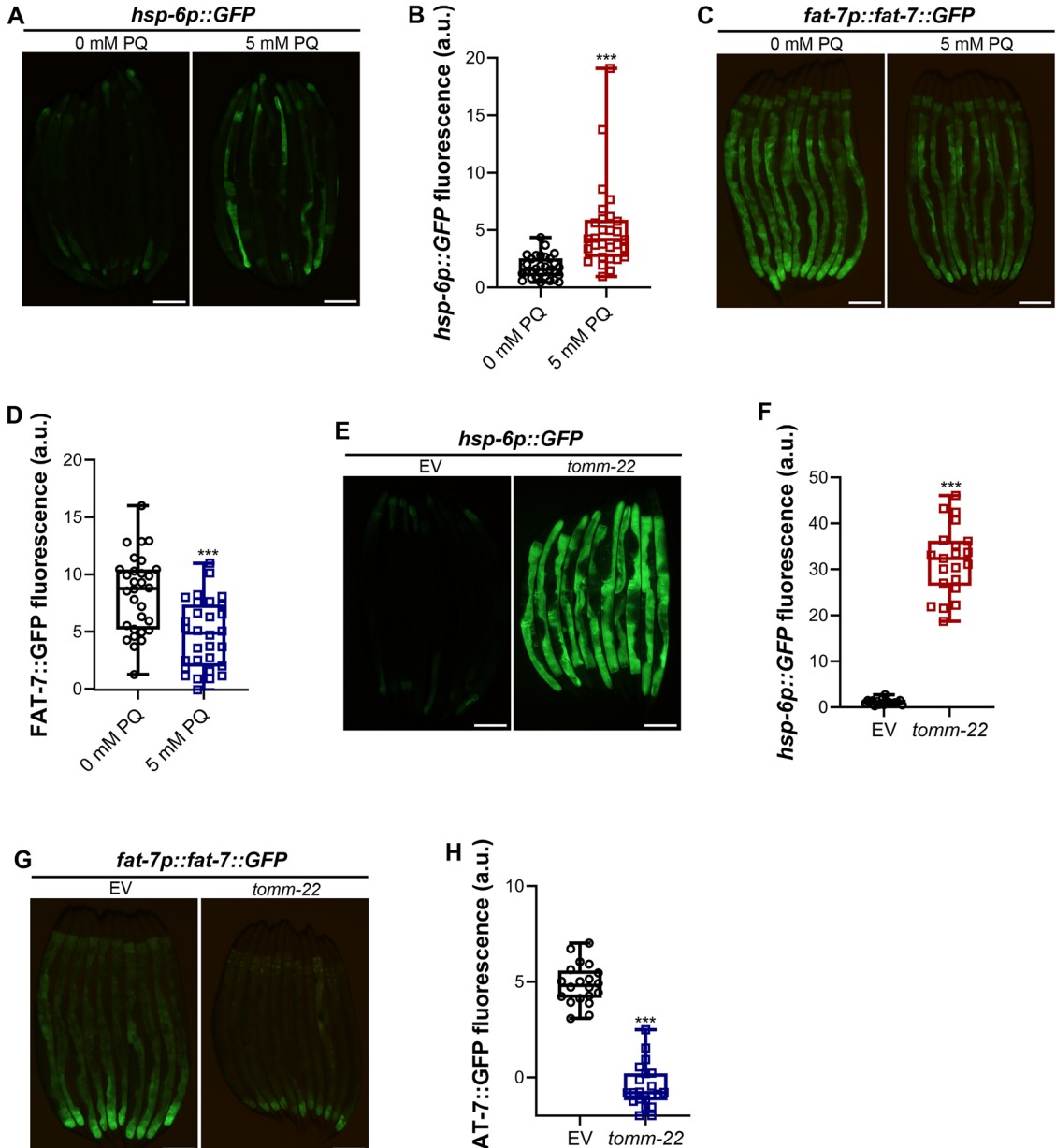

**Figure EV4. Mitochondrial stress suppresses *fat-7* expression.**

(A) Representative fluorescence images of *hsp-6p::GFP* worms exposed to *E. coli* BW25113 supplemented with 0 or 5 mM paraquat dichloride (PQ) for 24 h. Scale bar = 200 μm. (B) Quantification of GFP levels of *hsp-6p::GFP* worms exposed to *E. coli* BW25113 supplemented with 0 or 5 mM PQ for 24 h. ***P < 0.0001 via the *t* test (*n* = 29–30 worms each). (C) Representative fluorescence images of *fat-7p::fat-7::GFP* worms grown on *E. coli* BW25113 supplemented with 0 or 5 mM PQ for 24 h. Scale bar = 200 μm. (D) Quantification of GFP levels of *fat-7p::fat-7::GFP* worms grown on *E. coli* BW25113 supplemented with 0 or 5 mM PQ for 24 h. ***P < 0.0001 via the *t* test (*n* = 30 worms each). (E) Representative fluorescence images of *hsp-6p::GFP* worms grown on empty vector (EV) control or *tomm-22* RNAi. Scale bar = 200 μm. (F) Quantification of GFP levels of *hsp-6p::GFP* worms grown on EV control or *tomm-22* RNAi. ***p < 0.0001 via the *t* test (*n* = 19–21 worms each). (G) Representative fluorescence images of *fat-7p::fat-7::GFP* worms grown on EV control or *tomm-22* RNAi. Scale bar = 200 μm. (H) Quantification of GFP levels of *fat-7p::fat-7::GFP* worms grown on EV control or *tomm-22* RNAi. ***P < 0.0001 via the *t* test (*n* = 20–21 worms each). Data information: In the boxplots in (B, D, F, H), the central bands represent the median value, the boxes represent the upper and lower quartiles, and the whiskers represent the minimum and maximum values. Source data are available online for this figure.

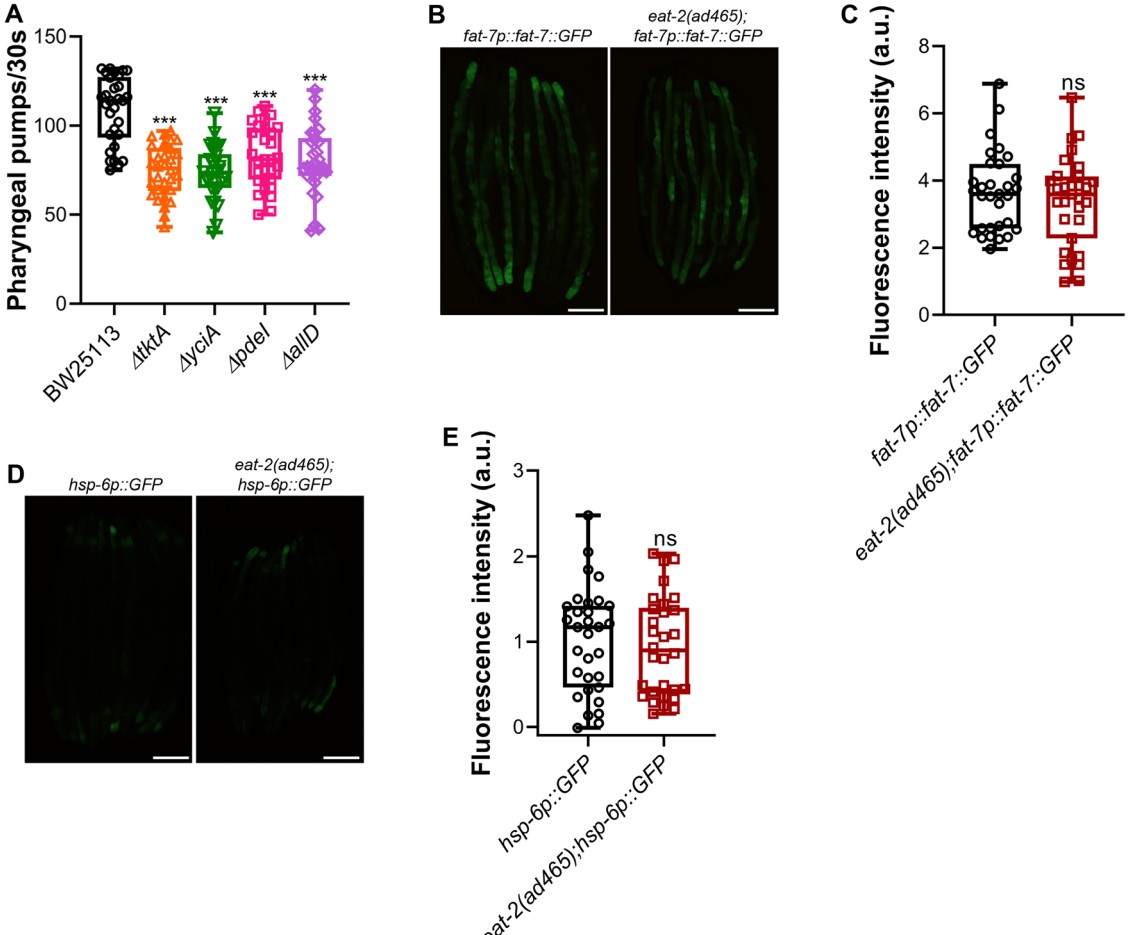

**Figure EV5. Reduced pharyngeal pumping does not cause mitochondrial stress.**

(A) Pharyngeal pumps per 30 s of 1-day-old adult N2 animals grown on *E. coli* BW25113 and FAT-7-suppressing diets at 20 °C. ***$P < 0.0001$ via the *t* test ($n = 30$ worms each). (B) Representative fluorescence images of *fat-7p::fat-7::GFP* and *eat-2(ad465);fat-7p::fat-7::GFP* worms grown on *E. coli* BW25113. Scale bar = 200 μm. (C) Quantification of GFP levels of *fat-7p::fat-7::GFP* and *eat-2(ad465);fat-7p::fat-7::GFP* worms grown on *E. coli* BW25113. $P = 0.4056$, ns, nonsignificant via the *t* test ($n = 31$ worms each). (D) Representative fluorescence images of *hsp-6p::GFP* and *eat-2(ad465);hsp-6p::GFP* worms grown on *E. coli* BW25113. Scale bar = 200 μm. (E) Quantification of GFP levels of *hsp-6p::GFP* and *eat-2(ad465);hsp-6p::*GFP worms grown on *E. coli* BW25113. $p = 0.6479$, ns, nonsignificant via the *t* test ($n = 30–31$ worms each). Data information: In the boxplots in (A, C, E), the central bands represent the median value, the boxes represent the upper and lower quartiles, and the whiskers represent the minimum and maximum values. Source data are available online for this figure.

