## [Peer Review File · The EMBO Journal]

Iron-deplete diet enhances *Caenorhabditis elegans* lifespan via oxidative stress response pathways

Priyanka Das, Ravi Ravi, and Jogender Singh

Corresponding author(s): Jogender Singh (jogender@iisermohali.ac.in)

Review Timeline:

Transfer from Review Commons:	7th May 25
Editorial Decision:	16th May 25
Revision Received:	25th Sep 25
Editorial Decision:	27th Oct 25
Revision Received:	28th Oct 25
Accepted:	28th Oct 25

Review
COMMONS

Editor: Hartmut Vodermaier

Transaction Report: This manuscript was transferred to The EMBO JOURNAL following peer review at Review Commons.

Review #1

1. Evidence, reproducibility and clarity:

Evidence, reproducibility and clarity (Required)

****Summary:****

In this paper, the authors perform a screen by feeding *C. elegans* different *E. coli* genetic mutants and examining the effect on the expression of fat-7, a stearyl-CoA 9-desaturase, which has been associated with longevity. They identify 26 *E. coli* strains that decrease fat-7 expression, all of which slow development and increase lifespan. RNA sequencing of worms treated with 4 of these strains identified genes involved in defense against oxidative stress among those genes that are commonly upregulated. Feeding *C. elegans* these 4 bacterial strains results in increased ROS and activation of the mitochondrial unfolded protein response, which appears to contribute to lifespan extension as these bacterial strains do not increase lifespan when the mitochondrial unfolded protein response transcription factor ATFS-1 is disrupted. Finally, the authors demonstrate a role for iron levels in mediating these phenotypes: iron supplementation inhibits the phenotypes caused by the identified bacterial strains, while iron chelation mimics these phenotypes.

****Major comments:****

The proposed model involves an increase in ROS levels activating the UPR_{mt} and then leading to lifespan extension. If the elevation in ROS levels is contributing then treatment with antioxidants should prevent UPR_{mt} activation and lifespan extension.

The authors suggest that iron depletion may disrupt iron-sulfur cluster proteins. The Rieske iron-sulfur protein ISP-1 from mitochondrial electron transport chain complex III has previously been associated with lifespan. Point mutations affecting the function of ISP-1 or RNAi decreasing the levels of ISP-1 both result in increased lifespan (PMID 20346072, 11709184). Thus, iron depletion may be increasing ROS, activating UPR_{mt} and increasing lifespan through decreasing ISP-1 levels.

All of the Kaplan-meier survival plots are missing statistical analyses. Please add p-values.

It would be helpful to include a model diagram of the proposed mechanisms in the main figures.

****Minor comments:****

Rather than "mutant diets" it would be more informative to call these "FAT-7-decreasing diets"

Is it surprising that none of the bacterial strains increased FAT-7 levels? Why do you think this is?

Page 5. "We hypothesized that diets reducing FAT-7 might elevate oleic acid levels". Since FAT-7 converts stearic acid to oleic acid, wouldn't decreasing FAT-7 levels decrease oleic acid levels and increase stearic acid levels?

Page 6. The authors cite Bennett et al. 2014 for the statement that "Activation of the UPR_{mt} has been associated with lifespan extension". This paper reaches the opposite conclusion "Activation of the mitochondrial unfolded protein response does not predict longevity in *Caenorhabditis elegans*". Also, in the Bennett paper and PMID 34585931, it is shown that constitutive activation of ATFS-1 decreases lifespan. Thus, the relationship between the UPR_{mt} and lifespan is not straightforward. These points should be mentioned.

Page 6. "Our transcriptomic analysis suggested elevated ROS". Rather than refer to gene expression, it would be better to refer to the ROS measurements that were performed.

The long-lived mitochondrial mutants *isp-1* and *nuo-6* have increased ROS, UPR_{mt} activation and increased lifespan. Multiple studies have examined gene expression in these long-lived mutant strains. How does gene expression in these mutants compare to worms treated with the FAT-7-decreasing *E. coli* mutants? While not necessary for this publication, it would be interesting to see whether the FAT-7-decreasing *E. coli* strains can increase *isp-1* and *nuo-6* lifespan.

SEK-1 is also involved in the p38-mediated innate immune signaling pathway, which has been shown to contribute to longevity in *C. elegans*. In fact, disruption of *sek-1* using RNAi decreased the lifespan of several long-lived mutant strains PMID 36514863.

Figure 2. Why were *cyoA* and *ycbk* chosen to show the full Kaplan-meier survival plot?

Figure 2, panel D. A better title may be "Mean Survival (Percent increase from control)"

While not necessary for this paper, it would be interesting to determine whether the FAT-7-decreasing E. coli strains alter resistance to oxidative stress.

Figure 4. It may be interesting to include a correlation plot comparing hsp-6::GFP fluorescence and lifespan. It looks like the magnitudes of increase for each phenotype are not correlated.

2. Significance:

Significance (Required)

Overall, this is an interesting paper and the experiments are rigorously performed. The bacterial screen was comprehensive and was followed up by careful mechanistic experiments. This paper will be of interest to researchers studying the biology of aging. A diagram of the working model of the underlying mechanisms would enhance the paper.

3. How much time do you estimate the authors will need to complete the suggested revisions:

Estimated time to Complete Revisions (Required)

(Decision Recommendation)

Between 1 and 3 months

4. Review Commons values the work of reviewers and encourages them to get credit for their work. Select 'Yes' below to register your reviewing activity at Web of Science Reviewer Recognition Service (formerly Publons); note that the content of your review will not be visible on Web of Science.

No

Review #2

1. Evidence, reproducibility and clarity:

Evidence, reproducibility and clarity (Required)

In this manuscript, Das et al. investigate how different bacterial mutants affect the lifespan of C. elegans. The authors screened a library of E. coli mutants using a fat-7 reporter and

identified 26 strains that reduce fat-7 expression, cause developmental delay, induce the mitochondrial unfolded protein response (using hsp-6 reporter), and increase worm lifespan. Among these, they focused on four strains and demonstrated that the effects of these mutants on developmental delay, fat-7 expression, and hsp-6 induction could be suppressed by iron supplementation. Furthermore, they showed that iron depletion alone is sufficient to induce fat-7 expression in worms. The lifespan extension observed in worms fed these mutant bacterial strains depends on SKN-1, SEK-1, and HLH-30.

Overall, this is a well-written manuscript that highlights the role of iron in regulating fat-7 expression. However, the findings from the initial screen do not significantly expand upon what is already known in the literature. Many of the identified hits overlap with those reported by Zhang et al. (2019), which also highlighted the role of iron in developmental delay and hsp-6 induction. While the lifespan data and the role of fat-7 are novel aspects of this study, the authors have not conducted detailed mechanistic investigations to address key questions, such as: 1) How does the deletion of these bacterial genes alter the metabolic state of the diet? 2) How do these metabolic changes influence fat-7 expression in worms? 3) How does the downregulation of fat-7 contribute to longevity? Addressing these points would strengthen the mechanistic insights of the study.

****Here are my detailed comments:****

1. Suppressing FAT-7 levels in *C. elegans* does not inherently increase lifespan. To directly attribute this effect to FAT-7, it would be important to attempt a rescue experiment to restore FAT-7 expression and assess whether the lifespan extension persists. Additionally, measuring oleic acid levels in these mutants would help determine whether a high-oleic-acid diet is suppressing FAT-7 expression. The role of oleic acid cannot be ruled out using fat-2 mutants (Fig. 3B), as fat-2 mutants accumulate oleic acid when fed WT bacteria, but this may not translate to endogenous oleic acid accumulation in conditions where FAT-7 is suppressed.
2. To understand the host-microbe interaction in this study, it is important to determine what specific changes in the bacteria contribute to the observed phenotypes in worms. Identifying these bacterial factors will provide a clearer picture of their role in influencing worms stress signaling and lifespan.
3. It is important to rule out any changes in food consumption in worms fed these bacterial mutants, as differences in feeding amount could attribute to the observed lifespan effects.
4. In figure 5A to 5G, please include the same-day controls to help clarify how iron supplementation effects these phenotypes relative to the control. For example, in Fig. 5F, it appears that iron extends the lifespan of worms fed the control diet. It would be clearer if

appropriate controls were included in all of these figures or summarized in a table to help understand the impact of iron.

5. How does iron depletion affect the levels of fat-7, and how does this contribute to the activation of the longevity pathways discussed in the manuscript.

****Minor comments****

1. Please include a detailed table of the lifespan data for all replicates as a supplementary table.

2. In the Methods section, specify at what stage the worms were exposed to iron and the iron chelator for the lifespan experiments.

3. Please clarify whether equal optical density (O.D.) of cells was seeded for both the WT and mutant strains, and mention if the mutants exhibit any growth defects.

2. Significance:

Significance (Required)

General Assessment: This study by Das et al. explores the impact of bacterial mutants on *C. elegans* lifespan. A key strength of the study is the identification of bacterial mutants that influence the expression of the gene encoding fatty acid desaturase (*fat-7*) and lifespan in *C. elegans*. Furthermore, the study highlights the role of iron in regulating *fat-7* expression, suggesting that iron imbalance may play a crucial role in modulating fatty acid metabolism. However, the study's main limitation is that it does not significantly extend the current understanding of the microbial modulation of host metabolism and aging, as many of the identified bacterial hits overlap with those previously reported in Zhang et al. (2019). The manuscript would benefit from more in-depth mechanistic exploration, especially with regard to how specific bacterial factors influence the metabolic state of the worms and how these changes ultimately modulate *fat-7* expression and longevity.

Advance: This study presents a conceptual advance by exploring the iron-dependent regulation of *fat-7* expression and lifespan in *C. elegans*, linking bacterial mutations with key longevity pathways (*SKN-1*, *SEK-1*, and *HLH-30*). The novelty lies in the direct investigation of the bacterial-induced changes in *fat-7* expression, though the role of iron in these mutants for development and induction of mito-UPR was previously shown in the literature. This study also adds to the growing body of work on *C. elegans* as a model for studying aging and host-microbe interactions, particularly in understanding how diet and microbial exposure affect metabolic processes and lifespan.

Audience: This research will primarily interest specialized audiences in aging research, microbiology, and metabolism, especially those focused on host-microbe interactions.

Keywords of my expertise: Host-microbe interactions, metabolism, system biology, C. elegans, aging.

3. How much time do you estimate the authors will need to complete the suggested revisions:

Estimated time to Complete Revisions (Required)

(Decision Recommendation)

More than 6 months

Yes

Revision Plan

Manuscript number: RC-2025-02952

Corresponding author(s): Jogender Singh

[The “revision plan” should delineate the revisions that authors intend to carry out in response to the points raised by the referees. It also provides the authors with the opportunity to explain their view of the paper and of the referee reports.]

The document is important for the editors of affiliate journals when they make a first decision on the transferred manuscript. It will also be useful to readers of the reprint and help them to obtain a balanced view of the paper.

*If you wish to submit a full revision, please use our "Full Revision" template. **It is important to use the appropriate template to clearly inform the editors of your intentions.**]*

1. General Statements [optional]

2. Description of the planned revisions

Insert here a point-by-point reply that explains what revisions, additional experimentations and analyses are planned to address the points raised by the referees.

Below, we have provided a point-by-point response to the reviewers' comments that explains our revision plans. All the comments are in blue font, while our responses are in black. We believe that we will be able to thoroughly address all the reviewers' concerns.

Reviewer #1 (Evidence, reproducibility and clarity (Required)):

Summary:

In this paper, the authors perform a screen by feeding *C. elegans* different *E. coli* genetic mutants and examining the effect on the expression of fat-7, a stearyl-CoA 9-desaturase, which has been associated with longevity. They identify 26 *E. coli* strains that decrease fat-7 expression, all of which slow development and increase lifespan. RNA sequencing of worms treated with 4 of these strains identified genes involved in defense against oxidative stress among those genes that are commonly upregulated. Feeding *C. elegans* these 4 bacterial strains results in increased ROS and activation of the mitochondrial unfolded protein response, which appears to contribute to lifespan extension as these bacterial strains do not increase lifespan when the mitochondrial unfolded protein response transcription factor ATFS-1 is disrupted. Finally, the authors demonstrate a role for iron levels in mediating these phenotypes: iron supplementation inhibits the phenotypes caused by the identified bacterial strains, while iron chelation mimics these phenotypes.

Revision Plan

Response: We thank the reviewer for an excellent summary of our work.

Major comments:

The proposed model involves an increase in ROS levels activating the UPRmt and then leading to lifespan extension. If the elevation in ROS levels is contributing then treatment with antioxidants should prevent UPRmt activation and lifespan extension.

Response: This is an excellent point. We will treat the FAT-7-suppressing diets with antioxidants and observe the effect on *C. elegans* UPRmt activation and lifespan.

The authors suggest that iron depletion may disrupt iron-sulfur cluster proteins. The Rieske iron-sulfur protein ISP-1 from mitochondrial electron transport chain complex III has previously been associated with lifespan. Point mutations affecting the function of ISP-1 or RNAi decreasing the levels of ISP-1 both result in increased lifespan (PMID 20346072, 11709184). Thus, iron depletion may be increasing ROS, activating UPRmt and increasing lifespan through decreasing ISP-1 levels.

Response: The reviewer has raised an intriguing possibility that the increased lifespan on the FAT-7-suppressing diets could be because of perturbation of ISP-1 function. While ISP-1 levels may not be directly affected by the mutant diets, ISP-1 function might be perturbed on these diets as ISP-1 function requires iron-sulfur clusters. Therefore, we will study the lifespan of *isp-1(qm150)* mutant on the FAT-7-suppressing diets to explore whether the lifespan extension on these diets is ISP-1 dependent.

All of the Kaplan-meier survival plots are missing statistical analyses. Please add p-values.

Response: The p-values for all the survival plots are included in the respective figure legends.

It would be helpful to include a model diagram of the proposed mechanisms in the main figures.

Response: We will make a model diagram after completing the experiments suggested by the reviewers.

Minor comments:

Rather than "mutant diets" it would be more informative to call these "FAT-7-decreasing diets"

Response: We have changed "mutant diets" to "FAT-7-suppressing diets" throughout the manuscript.

Is it surprising that none of the bacterial strains increased FAT-7 levels? Why do you think this is?

Response: Yes, it was indeed surprising to find only bacterial strains that reduced FAT-7 levels and none that increased them. One possible explanation is that these bacterial mutants may not directly regulate *fat-7* expression. Instead, they might alter the overall dietary composition,

Revision Plan

which is known to influence *fat-7* levels. It appears that none of the tested mutants modified the diet in a manner that would lead to *fat-7* upregulation.

Page 5. "We hypothesized that diets reducing FAT-7 might elevate oleic acid levels". Since FAT-7 converts stearic acid to oleic acid, wouldn't decreasing FAT-7 levels decrease oleic acid levels and increase stearic acid levels?

Response: FAT-7 expression is regulated by a feedback mechanism and is sensitive to the fatty acid composition within host cells; elevated levels of unsaturated fatty acids, such as oleic acid, suppress FAT-7 expression. There are two possible ways bacterial mutants could lead to reduced FAT-7 levels: (1) by directly inhibiting FAT-7 expression, which would be expected to result in increased stearic acid levels; or (2) by supplying higher amounts of oleic acid through their composition, thereby suppressing FAT-7 expression via feedback regulation. We focused on the second possibility, as elevated oleic acid levels—like those seen with FAT-7-suppressing diets—are known to promote *C. elegans* lifespan. To avoid confusion, we have revised the statement to: "*We hypothesized that bacterial diets might reduce FAT-7 expression because they have elevated levels of oleic acid*".

Page 6. The authors cite Bennett et al. 2014 for the statement that "Activation of the UPRmt has been associated with lifespan extension". This paper reaches the opposite conclusion "Activation of the mitochondrial unfolded protein response does not predict longevity in *Caenorhabditis elegans*". Also, in the Bennett paper and PMID 34585931, it is shown that constitutive activation of ATFS-1 decreases lifespan. Thus, the relationship between the UPRmt and lifespan is not straightforward. These points should be mentioned.

Response: The reviewer has raised an important point. We have now included a paragraph in the discussion to highlight these points. The revised manuscript reads: "*All 26 FAT-7-suppressing diets identified in our study elevated hsp-6p::GFP expression and extended C. elegans lifespan. Although UPRmt activation and lifespan extension were consistently observed across these diets, there was no strong correlation between hsp-6p::GFP levels and the degree of lifespan extension. The role of the UPRmt in promoting longevity remains controversial (Bennett et al., 2014; Soo et al., 2021; Wu et al., 2018). For instance, gain-of-function mutations in atfs-1 have been shown to reduce lifespan (Bennett et al., 2014; Soo et al., 2021). However, a recent study demonstrated that mild UPRmt activation can extend lifespan, whereas strong activation has the opposite effect (Di Pede et al., 2025). These findings suggest that UPRmt contributes to longevity only under specific conditions and at specific activation levels. In our study, lifespan extension on FAT-7-suppressing diets was dependent on ATFS-1, indicating that UPRmt activation was necessary for this effect.*"

Page 6. "Our transcriptomic analysis suggested elevated ROS". Rather than refer to gene expression, it would be better to refer to the ROS measurements that were performed.

Revision Plan

Response: We have changed it to the following sentence: “*Our ROS measurement analysis suggested elevated ROS levels in worms fed FAT-7-suppressing diets.*”

The long-lived mitochondrial mutants *isp-1* and *nuo-6* have increased ROS, UPRmt activation and increased lifespan. Multiple studies have examined gene expression in these long-lived mutant strains. How does gene expression in these mutants compare to worms treated with the FAT-7-decreasing *E. coli* mutants? While not necessary for this publication, it would be interesting to see whether the FAT-7-decreasing *E. coli* strains can increase *isp-1* and *nuo-6* lifespan.

Response: We will compare the gene expression changes observed in *isp-1* and *nuo-6* mutants with the gene expression changes observed in worms exposed to FAT-7-suppressing diets. Additionally, we will examine the lifespan of *isp-1* mutants on the mutant diets. These data will be included in the revised manuscript.

SEK-1 is also involved in the p38-mediated innate immune signaling pathway, which has been shown to contribute to longevity in *C. elegans*. In fact, disruption of *sek-1* using RNAi decreased the lifespan of several long-lived mutant strains PMID 36514863.

Response: We thank the reviewer for highlighting this point. We have now added that the role of SEK-1 in regulating lifespan on FAT-7-suppressing diets could also be because of its role in innate immunity. The revised manuscript reads: “*Notably, SEK-1 also regulates innate immunity and is essential for the extended lifespan observed in several long-lived C. elegans mutants (Soo et al., 2023). Therefore, its effect on lifespan in response to FAT-7-suppressing diets may also stem from its role in innate immune regulation.*”

Figure 2. Why were *cyoA* and *ycbk* chosen to show the full Kaplan-meier survival plot?

Response: These were selected randomly to show the range of the lifespan phenotype observed.

Figure 2, panel D. A better title may be "Mean Survival (Percent increase from control)"

Response: We have made this change.

While not necessary for this paper, it would be interesting to determine whether the FAT-7-decreasing *E. coli* strains alter resistance to oxidative stress.

Response: We will study the survival of worms on these diets upon supplementation with paraquat.

Revision Plan

Figure 4. It may be interesting to include a correlation plot comparing *hsp-6::GFP* fluorescence and lifespan. It looks like the magnitudes of increase for each phenotype are not correlated.

Response: We have added a new Figure (Figure S4) to show the correlation between *hsp-6::GFP* fluorescence levels and percent change in mean lifespan. Indeed, there is no correlation between these phenotypes.

Reviewer #1 (Significance (Required)):

Overall, this is an interesting paper and the experiments are rigorously performed. The bacterial screen was comprehensive and was followed up by careful mechanistic experiments. This paper will be of interest to researchers studying the biology of aging. A diagram of the working model of the underlying mechanisms would enhance the paper.

Response: We thank the reviewer for highlighting the significance of the study. We will include a model in the revised manuscript.

Reviewer #2 (Evidence, reproducibility and clarity (Required)):

In this manuscript, Das et al. investigate how different bacterial mutants affect the lifespan of *C. elegans*. The authors screened a library of *E. coli* mutants using a *fat-7* reporter and identified 26 strains that reduce *fat-7* expression, cause developmental delay, induce the mitochondrial unfolded protein response (using *hsp-6* reporter), and increase worm lifespan. Among these, they focused on four strains and demonstrated that the effects of these mutants on developmental delay, *fat-7* expression, and *hsp-6* induction could be suppressed by iron supplementation. Furthermore, they showed that iron depletion alone is sufficient to induce *fat-7* expression in worms. The lifespan extension observed in worms fed these mutant bacterial strains depends on *SKN-1*, *SEK-1*, and *HLH-30*.

Overall, this is a well-written manuscript that highlights the role of iron in regulating *fat-7* expression. However, the findings from the initial screen do not significantly expand upon what is already known in the literature. Many of the identified hits overlap with those reported by Zhang et al. (2019), which also highlighted the role of iron in developmental delay and *hsp-6* induction. While the lifespan data and the role of *fat-7* are novel aspects of this study, the authors have not conducted detailed mechanistic investigations to address key questions, such as: 1) How does the deletion of these bacterial genes alter the metabolic state of the diet? 2) How do these metabolic changes influence *fat-7* expression in worms? 3) How does the downregulation of *fat-7* contribute to longevity? Addressing these points would strengthen the mechanistic insights of the study.

Response: We thank the reviewer for a thoughtful summary of our work and for the valuable feedback provided to improve the manuscript. We would like to emphasize that the screening conditions and objectives of our study were fundamentally different from those of Zhang et al. (2019). Furthermore, Zhang et al. (2019) did not investigate the effects of the bacterial mutants

Revision Plan

identified in their screens on *C. elegans* lifespan. Notably, the 26 bacterial mutants identified in our screen do not overlap with those reported in previous studies that examined bacterial strains promoting *C. elegans* longevity. As detailed below, we will address the points raised by the reviewer that will certainly strengthen the mechanistic insights of the study.

Here are my detailed comments:

1. Suppressing FAT-7 levels in *C. elegans* does not inherently increase lifespan. To directly attribute this effect to FAT-7, it would be important to attempt a rescue experiment to restore FAT-7 expression and assess whether the lifespan extension persists. Additionally, measuring oleic acid levels in these mutants would help determine whether a high-oleic-acid diet is suppressing FAT-7 expression. The role of oleic acid cannot be ruled out using *fat-2* mutants (Fig. 3B), as *fat-2* mutants accumulate oleic acid when fed WT bacteria, but this may not translate to endogenous oleic acid accumulation in conditions where FAT-7 is suppressed.

Response: We thank the reviewer for these useful suggestions. We will overexpress FAT-7 under a pan-tissue promoter (*eft-3*) and study lifespan on FAT-7-suppressing diets. Moreover, to explore whether oleic acid has any role in enhancing lifespan on FAT-7-suppressing diets, we will study the lifespan of worms on these diets upon supplementing with oleic acid along with wild-type bacterium control.

2. To understand the host-microbe interaction in this study, it is important to determine what specific changes in the bacteria contribute to the observed phenotypes in worms. Identifying these bacterial factors will provide a clearer picture of their role in influencing worms stress signaling and lifespan.

Response: The phenotypes observed in *C. elegans* across all the identified bacterial mutants are remarkably consistent, including increased UPR^{mt} activation, reduced FAT-7 levels, delayed development, and extended lifespan. This consistency suggests that a common underlying factor is driving these effects. Although the bacterial mutants appear genetically diverse, gene expression data from *C. elegans*, along with comparisons to the findings of Zhang et al. (2019), indicate that elevated levels of reactive oxygen species (ROS) may represent this shared factor. These results suggest that bacterial ROS play a central role in mediating the host-microbe interactions underlying the observed phenotypes. To further support this hypothesis, we will directly measure ROS levels in the identified bacterial mutants. Additionally, we will test whether antioxidant treatment can suppress the *C. elegans* phenotypes, thereby establishing a causal role for bacterial ROS.

3. It is important to rule out any changes in food consumption in worms fed these bacterial mutants, as differences in feeding amount could attribute to the observed lifespan effects.

Response: We will carry out pharyngeal pumping rate measurements to study whether there is any difference in food consumption in worms fed these bacterial mutants.

Revision Plan

4. In figure 5A to 5G, please include the same-day controls to help clarify how iron supplementation affects these phenotypes relative to the control. For example, in Fig. 5F, it appears that iron extends the lifespan of worms fed the control diet. It would be clearer if appropriate controls were included in all of these figures or summarized in a table to help understand the impact of iron.

Response: We will include these controls in the revised manuscript.

5. How does iron depletion affect the levels of *fat-7*, and how does this contribute to the activation of the longevity pathways discussed in the manuscript.

Response: This is an intriguing question. There are at least two possible explanations: (1) oxidative stress may directly downregulate *fat-7* expression, and (2) iron depletion could reduce ferroptosis, which in turn may influence fatty acid metabolism. In the revised manuscript, we will include data on how oxidative stress affects FAT-7 expression.

Minor comments

1. Please include a detailed table of the lifespan data for all replicates as a supplementary table.

Response: We have included the details of survival curves for all the data in the new Table S2.

2. In the Methods section, specify at what stage the worms were exposed to iron and the iron chelator for the lifespan experiments.

Response: The L1-synchronized worms were exposed to iron and iron chelator plates and allowed to develop till the late L4 stage before being transferred to lifespan assay plates that also contained the respective supplements. This information is now included in the Methods section.

3. Please clarify whether equal optical density (O.D.) of cells was seeded for both the WT and mutant strains, and mention if the mutants exhibit any growth defects.

Response: We have examined the growth of the bacterial mutants and found that they do not exhibit growth defects. Therefore, for all the assays, NGM plates were seeded with saturated cultures of all the bacterial strains. We have now included the growth curves data in the manuscript (Figure S4).

Reviewer #2 (Significance (Required)):

Significance

General Assessment: This study by Das et al. explores the impact of bacterial mutants on *C. elegans* lifespan. A key strength of the study is the identification of bacterial mutants that influence the expression of the gene encoding fatty acid desaturase (*fat-7*) and lifespan in *C.*

Revision Plan

elegans. Furthermore, the study highlights the role of iron in regulating fat-7 expression, suggesting that iron imbalance may play a crucial role in modulating fatty acid metabolism. However, the study's main limitation is that it does not significantly extend the current understanding of the microbial modulation of host metabolism and aging, as many of the identified bacterial hits overlap with those previously reported in Zhang et al. (2019). The manuscript would benefit from more in-depth mechanistic exploration, especially with regard to how specific bacterial factors influence the metabolic state of the worms and how these changes ultimately modulate fat-7 expression and longevity.

Response: We thank the reviewer for highlighting the significance of our study. Once again, we would like to emphasize that the screening conditions and objectives of our study differed fundamentally from those of Zhang et al. (2019). Furthermore, Zhang et al. did not investigate the impact of the bacterial mutants identified in their screen on *C. elegans* lifespan. As outlined above, we will address the reviewer's comments, which will undoubtedly strengthen the mechanistic insights of our study.

Advance: This study presents a conceptual advance by exploring the iron-dependent regulation of fat-7 expression and lifespan in *C. elegans*, linking bacterial mutations with key longevity pathways (SKN-1, SEK-1, and HLH-30). The novelty lies in the direct investigation of the bacterial-induced changes in fat-7 expression, though the role of iron in these mutants for development and induction of mito-UPR was previously shown in the literature. This study also adds to the growing body of work on *C. elegans* as a model for studying aging and host-microbe interactions, particularly in understanding how diet and microbial exposure affect metabolic processes and lifespan.

Response: We thank the reviewer for highlighting the advancement made by our study.

Audience: This research will primarily interest specialized audiences in aging research, microbiology, and metabolism, especially those focused on host-microbe interactions.
Keywords of my expertise: Host-microbe interactions, metabolism, system biology, *C. elegans*, aging.

3. Description of the revisions that have already been incorporated in the transferred manuscript

Please insert a point-by-point reply describing the revisions that were already carried out and included in the transferred manuscript. If no revisions have been carried out yet, please leave this section empty.

Below, we include all the points that have already been addressed and included in the manuscript. The changes made in the manuscript in response to these points are highlighted. These points are also included in section 2 (Description of the planned revisions). All the comments are in blue font, while our responses are in black.

Revision Plan

Reviewer #1 (Evidence, reproducibility and clarity (Required)):

All of the Kaplan-meier survival plots are missing statistical analyses. Please add p-values.

Response: The p-values for all the survival plots are included in the respective figure legends.

Minor comments:

Rather than "mutant diets" it would be more informative to call these "FAT-7-decreasing diets"

Response: We have changed "mutant diets" to "FAT-7-suppressing diets" throughout the manuscript.

Is it surprising that none of the bacterial strains increased FAT-7 levels? Why do you think this is?

Response: Yes, it was indeed surprising to find only bacterial strains that reduced FAT-7 levels and none that increased them. One possible explanation is that these bacterial mutants may not directly regulate *fat-7* expression. Instead, they might alter the overall dietary composition, which is known to influence *fat-7* levels. It appears that none of the tested mutants modified the diet in a manner that would lead to *fat-7* upregulation.

Page 5. "We hypothesized that diets reducing FAT-7 might elevate oleic acid levels". Since FAT-7 converts stearic acid to oleic acid, wouldn't decreasing FAT-7 levels decrease oleic acid levels and increase stearic acid levels?

Response: FAT-7 expression is regulated by a feedback mechanism and is sensitive to the fatty acid composition within host cells; elevated levels of unsaturated fatty acids, such as oleic acid, suppress FAT-7 expression. There are two possible ways bacterial mutants could lead to reduced FAT-7 levels: (1) by directly inhibiting FAT-7 expression, which would be expected to result in increased stearic acid levels; or (2) by supplying higher amounts of oleic acid through their composition, thereby suppressing FAT-7 expression via feedback regulation. We focused on the second possibility, as elevated oleic acid levels—like those seen with FAT-7-suppressing diets—are known to promote *C. elegans* lifespan. To avoid confusion, we have revised the statement to: "*We hypothesized that bacterial diets might reduce FAT-7 expression because they have elevated levels of oleic acid*".

Page 6. The authors cite Bennett et al. 2014 for the statement that "Activation of the UPRmt has been associated with lifespan extension". This paper reaches the opposite conclusion "Activation of the mitochondrial unfolded protein response does not predict longevity in *Caenorhabditis elegans*". Also, in the Bennett paper and PMID 34585931, it is shown that

Revision Plan

constitutive activation of ATFS-1 decreases lifespan. Thus, the relationship between the UPR_{mt} and lifespan is not straightforward. These points should be mentioned.

Response: The reviewer has raised an important point. We have now included a paragraph in the discussion to highlight these points. The revised manuscript reads: “All 26 FAT-7-suppressing diets identified in our study elevated hsp-6p::GFP expression and extended *C. elegans* lifespan. Although UPR_{mt} activation and lifespan extension were consistently observed across these diets, there was no strong correlation between hsp-6p::GFP levels and the degree of lifespan extension. The role of the UPR_{mt} in promoting longevity remains controversial (Bennett et al., 2014; Soo et al., 2021; Wu et al., 2018). For instance, gain-of-function mutations in *atfs-1* have been shown to reduce lifespan (Bennett et al., 2014; Soo et al., 2021). However, a recent study demonstrated that mild UPR_{mt} activation can extend lifespan, whereas strong activation has the opposite effect (Di Pede et al., 2025). These findings suggest that UPR_{mt} contributes to longevity only under specific conditions and at specific activation levels. In our study, lifespan extension on FAT-7-suppressing diets was dependent on ATFS-1, indicating that UPR_{mt} activation was necessary for this effect.”

Page 6. "Our transcriptomic analysis suggested elevated ROS". Rather than refer to gene expression, it would be better to refer to the ROS measurements that were performed.

Response: We have changed it to the following sentence: “Our ROS measurement analysis suggested elevated ROS levels in worms fed FAT-7-suppressing diets.”

SEK-1 is also involved in the p38-mediated innate immune signaling pathway, which has been shown to contribute to longevity in *C. elegans*. In fact, disruption of *sek-1* using RNAi decreased the lifespan of several long-lived mutant strains PMID 36514863.

Response: We thank the reviewer for highlighting this point. We have now added that the role of SEK-1 in regulating lifespan on FAT-7-suppressing diets could also be because of its role in innate immunity. The revised manuscript reads: “Notably, *SEK-1* also regulates innate immunity and is essential for the extended lifespan observed in several long-lived *C. elegans* mutants (Soo et al., 2023). Therefore, its effect on lifespan in response to FAT-7-suppressing diets may also stem from its role in innate immune regulation.”

Figure 2. Why were *cyoA* and *ycbk* chosen to show the full Kaplan-meier survival plot?

Response: These were selected randomly to show the range of the lifespan phenotype observed.

Figure 2, panel D. A better title may be "Mean Survival (Percent increase from control)"

Response: We have made this change.

Revision Plan

Figure 4. It may be interesting to include a correlation plot comparing *hsp-6::GFP* fluorescence and lifespan. It looks like the magnitudes of increase for each phenotype are not correlated.

Response: We have added a new Figure (Figure S4) to show the correlation between *hsp-6::GFP* fluorescence levels and percent change in mean lifespan. Indeed, there is no correlation between these phenotypes.

Reviewer #2 (Evidence, reproducibility and clarity (Required)):

Minor comments

1. Please include a detailed table of the lifespan data for all replicates as a supplementary table.

Response: We have included the details of survival curves for all the data in the new Table S2.

2. In the Methods section, specify at what stage the worms were exposed to iron and the iron chelator for the lifespan experiments.

Response: The L1-synchronized worms were exposed to iron and iron chelator plates and allowed to develop till the late L4 stage before being transferred to lifespan assay plates that also contained the respective supplements. This information is now included in the Methods section.

3. Please clarify whether equal optical density (O.D.) of cells was seeded for both the WT and mutant strains, and mention if the mutants exhibit any growth defects.

Response: We have examined the growth of the bacterial mutants and found that they do not exhibit growth defects. Therefore, for all the assays, NGM plates were seeded with saturated cultures of all the bacterial strains. We have now included the growth curves data in the manuscript (Figure S4).

4. Description of analyses that authors prefer not to carry out

Please include a point-by-point response explaining why some of the requested data or additional analyses might not be necessary or cannot be provided within the scope of a revision. This can be due to time or resource limitations or in case of disagreement about the necessity of such additional data given the scope of the study. Please leave empty if not applicable.

Dr. Jogender Singh
Indian Institute of Science Education and Research Mohali
Biological Sciences
3F2, AB2, IISER Mohali
Mohali, Punjab 140306
India

16th May 2025

Re: EMBOJ-2025-121287-T
Iron-deplete diet enhances *Caenorhabditis elegans* lifespan via oxidative stress response pathways

Dear Dr. Singh,

Thank you for transferring your manuscript on gut microbe modulation of *C. elegans* lifespan from Review Commons to The EMBO Journal. I have now had the chance to read your study, as well as to go through the referee reports and your responses to them. I appreciate the question and approach as well as the comprehensive nature of the study, and I realize that your plans for addressing the overall constructive referee comments appear reasonable. We would therefore be happy to pursue a revised manuscript further for EMBO Journal publication, assuming that the planned experiments will provide conclusive answers to the reviewers' key queries. In this light, I am herewith inviting you to prepare and submit a new version revised along the lines suggested in your response letter.

When preparing a revised manuscript, please try to adhere to the guidelines listed below and in our Guide to Authors as closely as possible, as this should greatly facilitate our assessment at the time of resubmission - in particular regarding the completion of our author checklist, the inclusion of editable text files and individual figures files, and the conversion of "supplemental" material into Expanded View and/or Appendix content. Please also note that it is our policy to allow only a single round of (major) revision, making it important to comprehensively answer all criticisms at this point - if this should require more time than our standard three-months deadline, I would be happy to discuss an extension of the revision time, during which our 'scooping protection' (meaning that competing work appearing elsewhere in the meantime will not affect our considerations of your study) would of course remain valid. Please do not hesitate to contact me should you have any further questions at this stage.

Thank you again for the opportunity to consider this study for The EMBO Journal. I look forward to receiving your revision.

With kind regards,

Hartmut

- a point-by-point response to the referees' comments, with a detailed description of the changes made (as a word file).
- a word file of the manuscript text.

- individual production quality figure files (one file per figure)
- a complete author checklist, which you can download from our author guidelines (<https://www.embopress.org/page/journal/14602075/authorguide>).
- Expanded View files (replacing Supplementary Information)

We realize that it is difficult to revise to a specific deadline. In the interest of protecting the conceptual advance provided by the work, we recommend a revision within 3 months (14th Aug 2025). Please discuss the revision progress ahead of this time with the editor if you require more time to complete the revisions. Use the link below to submit your revision:

Link Not Available

Rev_Com_number: RC-2025-02952
New_manu_number: EMBOJ-2025-121287-T
Corr_author: Singh
Title: Iron-deplete diet enhances *Caenorhabditis elegans* lifespan via oxidative stress response pathways

Full Revision

Manuscript number: RC-2025-02952; *EMBO J.* manuscript number: EMBOJ-2025-121287-T

Corresponding author(s): Jogender Singh

[Please use this template only if the submitted manuscript should be considered by the affiliate journal as a full revision in response to the points raised by the reviewers.]

*If you wish to submit a preliminary revision with a revision plan, please use our "Revision Plan" template. **It is important to use the appropriate template to clearly inform the editors of your intentions.**]*

1. General Statements [optional]

This section is optional. Insert here any general statements you wish to make about the goal of the study or about the reviews.

We sincerely thank both reviewers for their constructive and positive comments, which have greatly helped improve the quality of our manuscript. In response, we performed several new experiments to address all concerns raised. As a result, the revised manuscript now includes important additions, such as the demonstration that the antioxidant NAC rescues all phenotypes associated with the bacterial mutant diets and that mitochondrial stress leads to the suppression of FAT-7 expression. These findings substantially strengthen the manuscript. We are grateful to the reviewers for their valuable feedback and hope that the revised version will be considered suitable for publication.

This section is mandatory. Please insert a point-by-point reply describing the revisions that were already carried out and included in the transferred manuscript.

Below, we have provided a point-by-point response to the reviewers' comments. All the comments are in blue font, while our responses are in black.

Reviewer #1 (Evidence, reproducibility and clarity (Required)):

Summary:

In this paper, the authors perform a screen by feeding *C. elegans* different *E. coli* genetic mutants and examining the effect on the expression of fat-7, a stearoyl-CoA 9-desaturase, which has been associated with longevity. They identify 26 *E. coli* strains that decrease fat-7 expression, all of which slow development and increase lifespan. RNA sequencing of worms treated with 4 of these strains identified genes involved in defense against oxidative stress among those genes that are commonly upregulated. Feeding *C. elegans* these 4 bacterial

strains results in increased ROS and activation of the mitochondrial unfolded protein response, which appears to contribute to lifespan extension as these bacterial strains do not increase lifespan when the mitochondrial unfolded protein response transcription factor ATFS-1 is disrupted. Finally, the authors demonstrate a role for iron levels in mediating these phenotypes: iron supplementation inhibits the phenotypes caused by the identified bacterial strains, while iron chelation mimics these phenotypes.

Response: We thank the reviewer for an excellent summary of our work.

Major comments:

The proposed model involves an increase in ROS levels activating the UPR_{mt} and then leading to lifespan extension. If the elevation in ROS levels is contributing then treatment with antioxidants should prevent UPR_{mt} activation and lifespan extension.

Response: We thank the reviewer for this excellent suggestion. To address it, we performed experiments with N-acetylcysteine (NAC) supplementation and found that NAC rescued all phenotypes observed on FAT-7-suppressing diets, including developmental delay, reduced FAT-7 expression, increased *hsp-6* expression, and extended lifespan. These data are included in the new Figure 5. The results are described on page 8, lines 249-257, as follows:

“Antioxidant supplementation rescues mutant diet-induced phenotypes

Because worms exhibited elevated ROS on the FAT-7-suppressing diets, we asked whether the associated phenotypes were driven by increased ROS levels. To test this, we supplemented the diets with the antioxidant NAC and examined the resulting phenotypes. NAC supplementation restored normal development in worms grown on FAT-7-suppressing diets (Fig. 5A). It also nearly completely rescued FAT-7::GFP expression (Fig. 5B, C) and significantly reduced hsp-6 expression (Fig. 5D, E). Importantly, NAC supplementation abolished the lifespan extension normally observed on FAT-7-suppressing diets (Fig. 5F). Together, these findings indicated that the phenotypes induced by FAT-7-suppressing diets are primarily mediated by elevated ROS.”

The authors suggest that iron depletion may disrupt iron-sulfur cluster proteins. The Rieske iron-sulfur protein ISP-1 from mitochondrial electron transport chain complex III has previously been associated with lifespan. Point mutations affecting the function of ISP-1 or RNAi decreasing the levels of ISP-1 both result in increased lifespan (PMID 20346072, 11709184). Thus, iron depletion may be increasing ROS, activating UPR_{mt} and increasing lifespan through decreasing ISP-1 levels.

Response: The reviewer has raised an intriguing possibility that the increased lifespan observed on FAT-7-suppressing diets could result from perturbation of ISP-1 function. Since ISP-1 requires iron-sulfur clusters for its activity, it is conceivable that FAT-7-suppressing diets extend lifespan by interfering with ISP-1 function. To test this, we examined the lifespan of *isp-1(qm150)* mutants on FAT-7-suppressing diets and found no further lifespan extension. These results are included on page 8, lines 228-231, as follows: *“Because isp-1(qm150) mutants also activate the UPR_{mt} and display extended lifespan (Wu et al., 2018), we asked whether their longevity pathway overlapped with that induced by FAT-7-suppressing diets. Indeed, lifespan extension was abolished in isp-1(qm150) mutants fed these diets (Fig. 4D).”*

All of the Kaplan-meier survival plots are missing statistical analyses. Please add p-values.

Response: The p -values for all the survival plots are included in the respective figure legends. We have also provided a table (Appendix Table S2) that includes all the replicate data for lifespan assays and their detailed statistical analysis.

It would be helpful to include a model diagram of the proposed mechanisms in the main figures.

Response: We thank the reviewer for the suggestion. We have included a model diagram in Figure 8 (Figure 8G).

Minor comments:

Rather than "mutant diets" it would be more informative to call these "FAT-7-decreasing diets"

Response: We have changed "mutant diets" to "FAT-7-suppressing diets" throughout the manuscript.

Is it surprising that none of the bacterial strains increased FAT-7 levels? Why do you think this is?

Response: Yes, it was indeed surprising that we identified only bacterial strains that reduced FAT-7 levels and none that increased them. However, our new results on the relationship between mitochondrial stress and FAT-7 expression (see below) provide an explanation. We observed that FAT-7 expression decreases under oxidative stress, whereas it increases when ROS levels are reduced, as seen with NAC supplementation. Thus, it is likely that none of the *E. coli* mutants lowered *C. elegans* ROS levels relative to wild-type *E. coli* BW25113.

Page 5. "We hypothesized that diets reducing FAT-7 might elevate oleic acid levels". Since FAT-7 converts stearic acid to oleic acid, wouldn't decreasing FAT-7 levels decrease oleic acid levels and increase stearic acid levels?

Response: FAT-7 expression is regulated by a feedback mechanism and is sensitive to the fatty acid composition within host cells; elevated levels of unsaturated fatty acids, such as oleic acid, suppress FAT-7 expression. There are two possible ways bacterial mutants could lead to reduced FAT-7 levels: (1) by directly inhibiting FAT-7 expression, which would be expected to result in increased stearic acid levels; or (2) by supplying higher amounts of oleic acid through their composition, thereby suppressing FAT-7 expression via feedback regulation. We focused on the second possibility, as elevated oleic acid levels—like those seen with FAT-7-suppressing diets—are known to promote *C. elegans* lifespan. To avoid confusion, we have revised the statement to: "We hypothesized that bacterial diets might reduce FAT-7 expression because they have elevated levels of oleic acid".

Page 6. The authors cite Bennett et al. 2014 for the statement that "Activation of the UPRmt has been associated with lifespan extension". This paper reaches the opposite conclusion "Activation of the mitochondrial unfolded protein response does not predict longevity in *Caenorhabditis elegans*". Also, in the Bennett paper and PMID 34585931, it is shown that constitutive activation of ATFS-1 decreases lifespan. Thus, the relationship between the UPRmt and lifespan is not straightforward. These points should be mentioned.

Response: The reviewer has raised an important point. We have now included a paragraph in the discussion to highlight these points. The revised manuscript reads (page 12, line 359-369): *"All 26 FAT-7-suppressing diets identified in our study elevated hsp-6p::GFP expression and extended C. elegans lifespan. Although UPRmt activation and lifespan extension were consistently observed across these diets, there was no strong correlation between hsp-6p::GFP levels and the degree of lifespan extension. The role of the UPRmt in promoting longevity remains controversial (Bennett et al., 2014; Soo et al., 2021; Wu et al., 2018). For instance, gain-of-function mutations in atfs-1 have been shown to reduce lifespan (Bennett et al., 2014; Soo et al., 2021). However, a recent study demonstrated that mild UPRmt activation can extend lifespan, whereas strong activation has the opposite effect (Di Pede et al., 2025). These findings suggest that UPRmt contributes to longevity only under specific conditions and at specific activation levels. In our study, lifespan extension on FAT-7-suppressing diets was dependent on ATFS-1, indicating that UPRmt activation was necessary for this effect."*

Page 6. "Our transcriptomic analysis suggested elevated ROS". Rather than refer to gene expression, it would be better to refer to the ROS measurements that were performed.

Response: We have changed it to the following sentence: *"Our ROS measurement analysis suggested elevated ROS levels in worms fed FAT-7-suppressing diets."*

The long-lived mitochondrial mutants *isp-1* and *nuo-6* have increased ROS, UPRmt activation and increased lifespan. Multiple studies have examined gene expression in these long-lived mutant strains. How does gene expression in these mutants compare to worms treated with the FAT-7-decreasing *E. coli* mutants? While not necessary for this publication, it would be interesting to see whether the FAT-7-decreasing *E. coli* strains can increase *isp-1* and *nuo-6* lifespan.

Response: We have compared the gene expression changes observed in *isp-1* and *nuo-6* mutants with the gene expression changes observed in worms exposed to FAT-7-suppressing diets, and found a significant overlap. These results are presented on page 7, lines 194-199, as follows: *"Mutations in the mitochondrial genes nuo-6 and isp-1, which encode subunits of complex I and III of the mitochondrial respiratory chain, respectively, are known to increase superoxide levels (Yang and Hekimi, 2010). A comparison of the genes upregulated on FAT-7-suppressing diets with those induced in nuo-6 and isp-1 partial loss-of-function mutants*

revealed significant overlap (Fig. 3E, F), supporting the notion that worms fed on FAT-7-suppressing diets experience elevated ROS.”

Additionally, we also examined the lifespan of *isp-1(qm150)* mutants on the FAT-7-suppressing diets and observed no further extension of lifespan in these worms. These results are presented on page 8, lines 228-231, as follows: “Because *isp-1(qm150)* mutants also activate the UPRmt and display extended lifespan (Wu et al., 2018), we asked whether their longevity pathway overlapped with that induced by FAT-7-suppressing diets. Indeed, lifespan extension was abolished in *isp-1(qm150)* mutants fed these diets (Fig. 4D).”

SEK-1 is also involved in the p38-mediated innate immune signaling pathway, which has been shown to contribute to longevity in *C. elegans*. In fact, disruption of *sek-1* using RNAi decreased the lifespan of several long-lived mutant strains PMID 36514863.

Response: We thank the reviewer for highlighting this point. We have now added that the role of SEK-1 in regulating lifespan on FAT-7-suppressing diets could also be because of its role in innate immunity. The revised manuscript reads (page 11, lines 326-329): “Notably, *SEK-1* also regulates innate immunity and is essential for the extended lifespan observed in several long-lived *C. elegans* mutants (Soo et al., 2023). Therefore, its effect on lifespan in response to FAT-7-suppressing diets may also stem from its role in innate immune regulation.”

Figure 2. Why were *cyoA* and *ycbk* chosen to show the full Kaplan-meier survival plot?

Response: These were selected randomly to show the range of the lifespan phenotype observed.

Figure 2, panel D. A better title may be “Mean Survival (Percent increase from control)”

Response: We have made this change.

While not necessary for this paper, it would be interesting to determine whether the FAT-7-decreasing *E. coli* strains alter resistance to oxidative stress.

Response: We thank the reviewer for this thoughtful suggestion. We are currently conducting an in-depth investigation into how the FAT-7-suppressing *E. coli* strains induce oxidative stress in *C. elegans*. As part of this work, we will also examine how these bacterial strains influence the worms’ resistance to oxidative stress.

Figure 4. It may be interesting to include a correlation plot comparing *hsp-6::GFP* fluorescence and lifespan. It looks like the magnitudes of increase for each phenotype are not correlated.

Response: We have added a new Figure panel (Appendix Figure S4A) to show the correlation between *hsp-6::GFP* fluorescence levels and percent change in mean lifespan. Indeed, there is no correlation between these phenotypes.

Reviewer #1 (Significance (Required)):

Overall, this is an interesting paper and the experiments are rigorously performed. The bacterial screen was comprehensive and was followed up by careful mechanistic experiments. This paper will be of interest to researchers studying the biology of aging. A diagram of the working model of the underlying mechanisms would enhance the paper.

Response: We thank the reviewer for highlighting the significance of the study and for providing useful comments that have improved the quality of the manuscript. We have included a model in the revised manuscript.

Reviewer #2 (Evidence, reproducibility and clarity (Required)):

In this manuscript, Das et al. investigate how different bacterial mutants affect the lifespan of *C. elegans*. The authors screened a library of *E. coli* mutants using a fat-7 reporter and identified 26 strains that reduce fat-7 expression, cause developmental delay, induce the mitochondrial unfolded protein response (using hsp-6 reporter), and increase worm lifespan. Among these, they focused on four strains and demonstrated that the effects of these mutants on developmental delay, fat-7 expression, and hsp-6 induction could be suppressed by iron supplementation. Furthermore, they showed that iron depletion alone is sufficient to induce fat-7 expression in worms. The lifespan extension observed in worms fed these mutant bacterial strains depends on SKN-1, SEK-1, and HLH-30.

Overall, this is a well-written manuscript that highlights the role of iron in regulating fat-7 expression. However, the findings from the initial screen do not significantly expand upon what is already known in the literature. Many of the identified hits overlap with those reported by Zhang et al. (2019), which also highlighted the role of iron in developmental delay and hsp-6 induction. While the lifespan data and the role of fat-7 are novel aspects of this study, the authors have not conducted detailed mechanistic investigations to address key questions, such as: 1) How does the deletion of these bacterial genes alter the metabolic state of the diet? 2) How do these metabolic changes influence fat-7 expression in worms? 3) How does the downregulation of fat-7 contribute to longevity? Addressing these points would strengthen the mechanistic insights of the study.

Response: We thank the reviewer for a thoughtful summary of our work and for the valuable feedback provided to improve the manuscript. We would like to emphasize that the screening conditions and objectives of our study were fundamentally different from those of Zhang et al. (2019). Furthermore, Zhang et al. (2019) did not investigate the effects of the bacterial mutants identified in their screens on *C. elegans* lifespan. Notably, the 26 bacterial mutants identified in our screen do not overlap with those reported in previous studies that examined bacterial strains promoting *C. elegans* longevity. As detailed below, we have addressed the points raised by the reviewer that have certainly strengthened the mechanistic insights of the study.

Here are my detailed comments:

1. Suppressing FAT-7 levels in *C. elegans* does not inherently increase lifespan. To directly attribute this effect to FAT-7, it would be important to attempt a rescue experiment to restore FAT-7 expression and assess whether the lifespan extension persists. Additionally, measuring oleic acid levels in these mutants would help determine whether a high-oleic-acid diet is suppressing FAT-7 expression. The role of oleic acid cannot be ruled out using *fat-2* mutants (Fig. 3B), as *fat-2* mutants accumulate oleic acid when fed WT bacteria, but this may not translate to endogenous oleic acid accumulation in conditions where FAT-7 is suppressed.

Response: We thank the reviewer for these helpful suggestions. We agree that the *fat-2* mutant data are only suggestive and do not fully exclude a role for oleic acid in the lifespan extension observed on the mutant diets. To further explore this possibility, we examined the effects of oleic acid supplementation on *C. elegans* lifespan. Oleic acid supplementation increased lifespan both on the control diet and on FAT-7-suppressing diets, indicating that the lifespan extension associated with the mutant diets is unlikely to be driven by oleic acid. These results are presented on page 6, lines 162-166, as follows: *“To further investigate the role of oleic acid in lifespan extension on FAT-7-suppressing diets, we examined the effects of oleic acid supplementation. As expected, oleic acid supplementation increased the lifespan of worms on the control diet (Fig. EV3A-D). Oleic acid also extended lifespan on the FAT-7-suppressing diets, suggesting that the observed lifespan extension under these conditions is unlikely to be driven by oleic acid.”*

We also overexpressed FAT-7 and examined its effect on lifespan on the mutant diets. These results are presented on page 6, lines 166-176, as follows: *“We next asked whether downregulation of FAT-7 itself was responsible for the extended lifespan on these diets. To address this, we overexpressed FAT-7 under an intestine-specific promoter. Although intestinal overexpression of FAT-7 has previously been reported to extend lifespan (S. Han et al., 2017), we did not observe increased lifespan upon FAT-7 overexpression in worms fed the control diet (Fig. EV3E-H). This discrepancy may reflect differences in experimental conditions, such as bacterial diets or transgene expression levels. Nonetheless, intestinal FAT-7 overexpression only partially reduced lifespan extension on the FAT-7-suppressing diets (Fig. EV3E-H), indicating that suppression of FAT-7 expression contributes only modestly to the observed phenotype. Collectively, these results suggested that, within our experimental framework, FAT-7 expression likely functions as an indirect proxy for lifespan regulation rather than a direct determinant.”*

Finally, we investigated the mechanism by which bacterial mutants suppress FAT-7 expression. We found that oxidative stress induced by bacterial mutants leads to suppression of FAT-7 expression in *C. elegans*. These results are described on page 8, lines 233-247, as follows: *“Given that the transcriptional profiles of worms fed FAT-7-suppressing diets significantly overlapped with *nuo-6* and *isp-1* loss-of-function mutants, we next asked whether these mitochondrial mutants also showed reduced *fat-7* expression. Indeed, transcriptomic data from multiple studies showed that *fat-7* is consistently downregulated in *nuo-6* and *isp-1* mutants (Park et al., 2020; Senchuk et al., 2018; Wu et al., 2018; Yee et al., 2014). This led us to hypothesize that mitochondrial stress more broadly downregulates *fat-7*. Supporting this,*

reanalysis of published datasets revealed reduced fat-7 expression in several mitochondrial mutants with activated UPRmt, including clk-1, cco-1, and hsp-6 (Fischer et al., 2014; Mao et al., 2019; Matilainen et al., 2017; Tian et al., 2016; Zhu et al., 2020). To confirm whether mitochondrial stress results in the downregulation of fat-7, we exposed the fat-7p::fat-7::GFP reporter strain to paraquat (PQ). While PQ treatment resulted in the upregulation of hsp-6p::GFP, it led to the downregulation of FAT-7::GFP levels (Fig. EV4A-D). Similarly, knockdown of tomm-22, which elicits UPRmt, also led to downregulation of FAT-7::GFP (Fig. EV4E-H). Together, these findings suggested that mitochondrial stress suppresses fat-7 expression and that the FAT-7 reporter may have functioned as an indirect indicator of mitochondrial stress in our Keio library screen.”

Collectively, our new data indicate that oleic acid is unlikely to be the dietary component responsible for lifespan extension on mutant diets, and direct regulation of FAT-7 expression is also unlikely to underlie the observed phenotypes. Rather, FAT-7 expression is sensitive to oxidative stress, and the FAT-7 reporter may have served as an indirect indicator of mitochondrial stress in our Keio library screen.

2. To understand the host-microbe interaction in this study, it is important to determine what specific changes in the bacteria contribute to the observed phenotypes in worms. Identifying these bacterial factors will provide a clearer picture of their role in influencing worms stress signaling and lifespan.

Response: The phenotypes observed in *C. elegans* across all the identified bacterial mutants are remarkably consistent, including increased UPRmt activation, reduced FAT-7 levels, delayed development, and extended lifespan. This consistency suggests that a common underlying factor is driving these effects. Although the bacterial mutants appear genetically diverse, gene expression data from *C. elegans*, along with comparisons to the findings of Zhang et al. (2019), indicate that elevated levels of reactive oxygen species (ROS) may represent this shared factor. These results suggest that bacterial ROS play a central role in mediating the host-microbe interactions underlying the observed phenotypes. To further establish whether bacterial ROS is the contributing factor that results in all the observed phenotypes, we supplemented mutant diets with the antioxidant NAC and observed that NAC reversed all the phenotypes observed on the mutant diets. These new results are presented in the new Figure 5 on page 8, lines 249-257, as follows: **“Antioxidant supplementation rescues mutant diet-induced phenotypes**

Because worms exhibited elevated ROS on the FAT-7-suppressing diets, we asked whether the associated phenotypes were driven by increased ROS levels. To test this, we supplemented the diets with the antioxidant NAC and examined the resulting phenotypes. NAC supplementation restored normal development in worms grown on FAT-7-suppressing diets (Fig. 5A). It also nearly completely rescued FAT-7::GFP expression (Fig. 5B, C) and significantly reduced hsp-6 expression (Fig. 5D, E). Importantly, NAC supplementation abolished the lifespan extension normally observed on FAT-7-suppressing diets (Fig. 5F). Together, these findings indicated that the phenotypes induced by FAT-7-suppressing diets are primarily mediated by elevated ROS.”

3. It is important to rule out any changes in food consumption in worms fed these bacterial mutants, as differences in feeding amount could attribute to the observed lifespan effects.

Response: We thank the reviewer for this valuable suggestion. We measured pharyngeal pumping rates on the mutant diets and found that worms exhibited a significant reduction in pumping. However, our analyses indicate that this decline is unlikely to account for the observed phenotypes. The results are presented on page 10, lines 292-302, as follows: *“We next investigated the mechanisms underlying lifespan extension in worms fed FAT-7-suppressing or iron-depleted diets. Because changes in food intake can influence lifespan, we first tested whether the mutant diets affected feeding behavior. Worms fed FAT-7-suppressing diets showed a significant reduction in pharyngeal pumping (Fig. EV5A). To determine whether reduced food intake accounted for the observed phenotypes, we examined eat-2(ad465) mutants, which display markedly reduced pharyngeal pumping (Avery, 1993). If decreased pumping were causal, eat-2 mutants should exhibit reduced fat-7 expression and elevated hsp-6 expression. However, eat-2 mutants showed neither phenotype (Fig. EV5B-E). Thus, reduced pharyngeal pumping was not the cause of the observed phenotypes but was more likely a consequence of elevated oxidative stress. Supporting this idea, mitochondrial mutants with increased oxidative stress are known to show reduced pumping (Jafari et al., 2015; Yee et al., 2014).”*

4. In figure 5A to 5G, please include the same-day controls to help clarify how iron supplementation effects these phenotypes relative to the control. For example, in Fig. 5F, it appears that iron extends the lifespan of worms fed the control diet. It would be clearer if appropriate controls were included in all of these figures or summarized in a table to help understand the impact of iron.

Response: We have conducted the ferric chloride supplementation experiments again, along with their same-day controls without ferric chloride supplementation. These data are included in the revised figure (now Figure 6).

5. How does iron depletion affect the levels of fat-7, and how does this contribute to the activation of the longevity pathways discussed in the manuscript.

Response: We thank the reviewer for raising this intriguing question. In response, we investigated the mechanism underlying the downregulation of FAT-7 on the mutant diets. In the revised manuscript, we demonstrate that mitochondrial stress, driven by elevated ROS, leads to reduced FAT-7 expression. Thus, FAT-7 expression likely served as an indirect indicator of mitochondrial stress in our Keio library screen.

Minor comments

1. Please include a detailed table of the lifespan data for all replicates as a supplementary table.

Response: We have included the details of survival curves for all the data in the new Appendix Table S2.

2. In the Methods section, specify at what stage the worms were exposed to iron and the iron chelator for the lifespan experiments.

Response: The L1-synchronized worms were exposed to iron and iron chelator plates and allowed to develop till the late L4 stage before being transferred to lifespan assay plates that also contained the respective supplements. This information is now included in the Methods section.

3. Please clarify whether equal optical density (O.D.) of cells was seeded for both the WT and mutant strains, and mention if the mutants exhibit any growth defects.

Response: We have examined the growth of the bacterial mutants and found that they do not exhibit growth defects. Therefore, for all the assays, NGM plates were seeded with saturated cultures of all the bacterial strains. We have now included the growth curves data in the manuscript (Appendix Figure S2).

Reviewer #2 (Significance (Required)):

Significance

General Assessment: This study by Das et al. explores the impact of bacterial mutants on *C. elegans* lifespan. A key strength of the study is the identification of bacterial mutants that influence the expression of the gene encoding fatty acid desaturase (*fat-7*) and lifespan in *C. elegans*. Furthermore, the study highlights the role of iron in regulating *fat-7* expression, suggesting that iron imbalance may play a crucial role in modulating fatty acid metabolism. However, the study's main limitation is that it does not significantly extend the current understanding of the microbial modulation of host metabolism and aging, as many of the identified bacterial hits overlap with those previously reported in Zhang et al. (2019). The manuscript would benefit from more in-depth mechanistic exploration, especially with regard to how specific bacterial factors influence the metabolic state of the worms and how these changes ultimately modulate *fat-7* expression and longevity.

Response: We thank the reviewer for highlighting the significance of our study. Once again, we would like to emphasize that the screening conditions and objectives of our study differed fundamentally from those of Zhang et al. (2019). Furthermore, Zhang et al. did not investigate the impact of the bacterial mutants identified in their screen on *C. elegans* lifespan. As outlined above, we have addressed the reviewer's comments, which have undoubtedly strengthened the mechanistic insights of our study.

Advance: This study presents a conceptual advance by exploring the iron-dependent regulation of *fat-7* expression and lifespan in *C. elegans*, linking bacterial mutations with key longevity

Full Revision

pathways (SKN-1, SEK-1, and HLH-30). The novelty lies in the direct investigation of the bacterial-induced changes in fat-7 expression, though the role of iron in these mutants for development and induction of mito-UPR was previously shown in the literature. This study also adds to the growing body of work on *C. elegans* as a model for studying aging and host-microbe interactions, particularly in understanding how diet and microbial exposure affect metabolic processes and lifespan.

Response: We thank the reviewer for highlighting the advancement made by our study.

Audience: This research will primarily interest specialized audiences in aging research, microbiology, and metabolism, especially those focused on host-microbe interactions.

Keywords of my expertise: Host-microbe interactions, metabolism, system biology, *C. elegans*, aging.

Dr. Jogender Singh
Indian Institute of Science Education and Research Mohali
Biological Sciences
3F2, AB2, IISER Mohali
Mohali, Punjab 140306
India

27th Oct 2025

Re: EMBOJ-2025-121287R

Iron-deplete diet enhances *Caenorhabditis elegans* lifespan via oxidative stress response pathways

Dear Jogender ,

Thank you for submitting your revised manuscript to The EMBO Journal. The two original Review Commons referees have now both assessed it once more, and were generally satisfied with the revisions. Referee 2 still notes a few minor presentational issues that should be incorporated during a final round of minor revision. In addition, please also address the following remaining editorial issues at this stage:

- Please double-check to make sure to all relevant funding information in the manuscript is congruent with the info entered into our submission system. Currently missing in the submission system are: the Council of Scientific & Industrial Research (CSIR), India; and IISER Mohali intramural funds
- Please carefully go through the reference list and make sure that each reference is complete with citation year, volume, and page/locator numbers - this information is currently missing for several of them - and that alternative DOI information is only included for pre-publication manuscripts that do not have a formal citation description yet.
- Please convert the separately uploaded "Appendix tables" into Expanded View material - Table S1 should become "Table EV1" and Tables S2-S4 should become "Dataset EV1-3", with call-outs in the text adjust accordingly. Also, please remove their listing and their legends from the Appendix PDF, and include the headers/legends for the EV Datasets instead on a separate tab of each respective XLSX spreadsheet.
- In the Appendix, please make sure to consistently use the nomenclature "Appendix Fig. S1, 2, 3...", also in the Figure legends.
- Finally, during routine pre-acceptance checks, our data editors have raised the following queries regarding figures, data, and legends, which I would ask you to address (ideally using the Track Changes option):
 1. Please note that the exact p-values have to be provided in the legends of figures 1B, 2D, 3H, 4B, 5C, E; 6C, E; 7C, E, K; EV4 B, D, F, H; EV5 A
 2. Please indicate the statistical test used for data analysis in the legends of figures 3D, S3 B
 3. Please note that the box plots need to be defined in terms of minima, maxima, centre, bounds of box and whiskers, and percentile in the legends of figures 1B, 3H, 4B, 5C, E; 6C, E; 7C, E; EV4 B, D, F, H; EV5 A, C, E
 4. Please note that information related to N is missing in the legends of figures 1B, S2
 5. Please note that the error bars are not defined in the legends of figures 1C, 5A, 6A, 7A, S2, S4 C

I am returning the manuscript to you for a final round of minor revision, to allow you to make these various modifications and to upload the revised files. Once we will have received them, we should be ready to proceed with formal acceptance and production of the manuscript.

With kind regards,

Hartmut

Revision to The EMBO Journal should be submitted online within 90 days, unless an extension has been requested and approved by the editor; please click on the link below to submit the revision online before 25th Jan 2026:

Link Not Available

Referee #1:

The authors have addressed all of my comments and concerns in a careful and thorough manner. I recommend acceptance of this manuscript.

Referee #2:

The revised manuscript by Das et al. is substantially improvement and effectively addresses all of my previous concerns. The study highlights the role of bacterial mutants in inducing ROS, which in turn reduces fat-7 expression and activates the mitochondrial UPR. This effect is attributed to reduced iron levels.

I have some minor suggestions for this manuscript:

1. Line 162 -176: Please give a clear explanation of why intestinal-specific rescue was used, and does the partial rescue by intestinal fat-7 suggest that other expression of fat-7 in other tissues also plays a role?
2. Test for the interaction between genotype and oleic acid treatment (e.g., using a Cox regression analysis) to determine whether the lifespan extension by oleic acid is additive or reduced in the bacterial mutants.
3. In the method section, please mention what developmental stage of worms was used for RNA-seq.
4. WT bacteria don't have the kan cassette, and as per the Methods section, WT was grown in LB while mutants were grown in the presence of antibiotics before seeding on NGM plates. Please clarify whether antibiotics were used for mutants when comparing the growth curve of WT vs mutants.

We thank the editor and reviewers for their interest in our study. We have now addressed all the remaining concerns. The quality of the manuscript has improved substantially through the review process, and we are grateful to both the reviewers and the editor for their constructive and insightful feedback.

Below, we provide a point-by-point response to the remaining issues.

Referee #1:

Comment: The authors have addressed all of my comments and concerns in a careful and thorough manner. I recommend acceptance of this manuscript.

Response: We sincerely thank the reviewer for the positive assessment and constructive feedback, which greatly contributed to improving the quality and clarity of our manuscript.

Referee #2:

Comment: The revised manuscript by Das et al. is substantially improvement and effectively addresses all of my previous concerns. The study highlights the role of bacterial mutants in inducing ROS, which in turn reduces fat-7 expression and activates the mitochondrial UPR. This effect is attributed to reduced iron levels.

Response: We sincerely thank the reviewer for the positive evaluation and insightful feedback, which have greatly contributed to improving the quality and clarity of the manuscript.

I have some minor suggestions for this manuscript:

Comment: 1. Line 162 -176: Please give a clear explanation of why intestinal-specific rescue was used, and does the partial rescue by intestinal fat-7 suggest that other expression of fat-7 in other tissues also plays a role?

Response: We have clarified the rationale for performing intestinal overexpression of FAT-7 in the revised manuscript. The updated text reads: “*To test this, we overexpressed FAT-7 to determine whether this manipulation could reverse the lifespan extension seen on the mutant diets. A previous study reported that intestinal overexpression of FAT-7 extends C. elegans lifespan (Han et al, 2017b). Moreover, FAT-7 expression is primarily observed in the intestine in the FAT-7::GFP reporter strain. Therefore, we overexpressed FAT-7 under an intestine-specific promoter. Unexpectedly, we did not observe increased lifespan upon FAT-7 overexpression in worms fed the control diet (Fig. EV3E-H). This discrepancy from Han et al, 2017b may reflect differences in experimental conditions, such as bacterial diets or transgene expression levels.*”

We agree with the reviewer that FAT-7 expression in other tissues may also contribute to the observed phenotypes, and investigating these possibilities will be an interesting direction for future studies.

Comment: 2. Test for the interaction between genotype and oleic acid treatment (e.g., using a Cox regression analysis) to determine whether the lifespan extension by oleic acid is additive or reduced in the bacterial mutants.

Response: As suggested by the reviewer, we performed a Cox regression analysis and found that oleic acid supplementation had widely ranging (negative to positive) effects on mutant diets. Overall, the data suggest that oleic acid is unlikely to be the primary factor that modulates *C. elegans* lifespan on the bacterial mutant diets.

Comment: 3. In the method section, please mention what developmental stage of worms was used for RNA-seq.

Response: We have now clarified that worms were grown to the day 1 adult stage before being used for RNA isolation.

Comment: 4. WT bacteria don't have the kan cassette, and as per the Methods section, WT was grown in LB while mutants were grown in the presence of antibiotics before seeding on NGM plates. Please clarify whether antibiotics were used for mutants when comparing the growth curve of WT vs mutants.

Response: No antibiotics were used for any of the bacterial strains during the growth curve assays. This is updated in the Methods.

Editorial issues:

Comment: - Please double-check to make sure to all relevant funding information in the manuscript is congruent with the info entered into our submission system. Currently missing in the submission system are: the Council of Scientific & Industrial Research (CSIR), India; and IISER Mohali intramural funds

Response: This information is updated.

Comment: - Please carefully go through the reference list and make sure that each reference is complete with citation year, volume, and page/locator numbers - this information is currently missing for several of them - and that alternative DOI information is only included for pre-publication manuscripts that do not have a formal citation description yet.

Response: We have reviewed and verified all references, and the missing citation details have been added.

Comment: - Please convert the separately uploaded "Appendix tables" into Expanded View material - Table S1 should become "Table EV1" and Tables S2-S4 should become "Dataset EV1-3", with call-outs in the text adjust accordingly. Also, please remove their listing and

their legends from the Appendix PDF, and include the headers/legends for the EV Datasets instead on a separate tab of each respective XLSX spreadsheet.

Response: These changes have been made.

Comment: - In the Appendix, please make sure to consistently use the nomenclature "Appendix Fig. S1, 2, 3...", also in the Figure legends.

Response: These changes have been made.

Comment: - Finally, during routine pre-acceptance checks, our data editors have raised the following queries regarding figures, data, and legends, which I would ask you to address (ideally using the Track Changes option):

1. Please note that the exact p -values have to be provided in the legends of figures 1B, 2D, 3H, 4B, 5C, E; 6C, E; 7C, E, K; EV4 B, D, F, H; EV5 A

Response: We have included the exact p -values in the revised manuscript. Significance values are reported to four decimal places; therefore, any p -value smaller than 0.0001 is denoted as $p < 0.0001$, while p -values greater than or equal to 0.0001 are presented as their exact values in the figure legends.

2. Please indicate the statistical test used for data analysis in the legends of figures 3D, S3 B

Response: This has been updated.

3. Please note that the box plots need to be defined in terms of minima, maxima, centre, bounds of box and whiskers, and percentile in the legends of figures 1B, 3H, 4B, 5C, E; 6C, E; 7C, E; EV4 B, D, F, H; EV5 A, C, E

Response: The box plots have been defined.

4. Please note that information related to N is missing in the legends of figures 1B, S2

Response: This has been updated.

5. Please note that the error bars are not defined in the legends of figures 1C, 5A, 6A, 7A, S2, S4 C

Response: This information has been provided.

In summary, we sincerely thank the editor and reviewers for their insightful suggestions, which have helped improve the quality of the manuscript. We believe that we have thoroughly addressed all remaining concerns and hope that the revised version will be considered suitable for publication in *The EMBO Journal*.